# Omni-DNA: A Genomic Model Supporting Sequence Understanding, Long-context, and Textual Annotation

**Zehui Li**[1,2,*], **Vallijah Subasri**[3,4], **Yifei Shen**[2], **Dongsheng Li**[2], **Wentao Gu**[2],
**Guy-Bart Stan**[1,†], **Yiren Zhao**[1,†], **Caihua Shan**[2,*]

[1]Imperial College London    [2]Microsoft Research
[3]Vector Institute    [4]University Health Network

[*]Correspondence to: {zl6222@ic.ac.uk, caihua.shan@microsoft.com}
[†]Co-corresponding authors: {g.stan@imperial.ac.uk, a.zhao@imperial.ac.uk}

## Abstract

The interpretation of genomic sequences is crucial for understanding biological processes. To handle the growing volume of DNA sequence data, Genomic Foundation Models (GFMs) have been developed by adapting architectures and training paradigms from Large Language Models (LLMs). Despite their remarkable performance in DNA sequence classification tasks, there remains a lack of systematic understanding regarding the pre-training and task-adaptation processes of GFMs. Moreover, existing GFMs cannot achieve state-of-the-art performance on both short and long-context tasks and lack multimodal abilities.

By revisiting pre-training architectures and post-training techniques, we propose OMNI-DNA, a family of models spanning 20M to 1.1B parameters that supports sequence understanding, long-context genomic reasoning, and natural-language annotation. Omni-DNA establishes new state-of-the-art results on 18 of 26 evaluations drawn from Nucleotide Transformer and Genomic Benchmarks. When jointly fine-tuning on biologically related tasks, Omni-DNA consistently outperforms existing models and demonstrates multi-tasking abilities. Furthermore, we introduce SEQ-PACK, an adaptive compression mechanism that enables efficient long-context modeling by summarizing historical tokens through position-aware learnable sampling. This allows transformer-based models to process ultra-long genomic sequences with minimal memory and computational overhead. Leveraging SEQPACK, Omni-DNA excels at enhancer–target interaction prediction, capturing distal regulatory effects over 450kbp. Finally, we present SEQ2FUNC, a newly constructed dataset that empowers Omni-DNA to generate accurate and functionally meaningful interpretations of DNA sequences, opening new avenues for genomic analysis and discovery.

## 1 Introduction

As genomic data has been expanding exponentially over recent decades [19], Genomic Foundation Models (GFMs) have emerged as an essential paradigm to model genomic sequences and complete various tasks, including variant effect prediction [21, 44], sequence generation [39], and functional classification [10]. Most GFMs follow a "pretrain + task-specific finetune" paradigm: first pretraining on unlabeled DNA sequences, and finetuning for each specific task.

While GFMs have demonstrated utility on specific genomic tasks [13, 47], they still exhibit important limitations: **(i) Limited optimization of architectural components for genomic tasks:** Prior work has explored design choices including tokenization (DNABERT [17] vs. DNABERT-2 [49]), positional encodings (NT vs. NTv2 [10]), and genomic-specific modules (GPN-MSA [5], Caduceus [30]),

but a more comprehensive, task-specific search of architectures and pretraining strategies tailored to genomic inductive biases is still lacking. **(ii) Limited exploration of multi-task synergies:** Many approaches fine-tune on a single downstream task, missing the opportunity to exploit shared biological signals and achieve cross-task performance gains, despite evidence of such benefits in genomic assay prediction (e.g., AlphaGenome [3], Enformer [2]). **(iii) Insufficient multimodal capability:** Most existing GFMs restrict vocabularies to nucleotide base pairs, predefined labels, or learned subwords via BPE tokenization, limiting their ability to handle tasks that require mapping DNA sequences to functional annotations in natural language. While ChatNT can process joint (DNA, text) inputs, it employs separate tokenizers and a two-stage architecture consisting of NTv2 [10] as the DNA encoder and Vicuna-7B [38] as the decoder, resulting in less unified multimodal integration. **(iv) Context-length adaptivity challenges:** Transformer-based models remain constrained by the quadratic cost of self-attention, limiting their scalability to long-context genomic tasks such as enhancer–target prediction.

To bring DNA sequence modeling closer to the breakthroughs that large language models have enabled in natural language processing, we comprehensively explored the design space of GFMs, from pre-training architectures and objectives to post-training techniques, and developed several targeted improvements. To this end, we propose Omni-DNA (Figure 1), a family of GFMs ranging from 20M to 1.1B parameters that enables multitask learning, long-context genomic reasoning, and natural-language annotation. We then outline how Omni-DNA addresses these challenges.

**Optimized backbone (Section 2).** Early GFMs such as DNABERT [17], DNABERT-2 [49] and Nucleotide Transformer [10] rely on masked language modeling within a BERT framework; recent models, including HyenaDNA [24], BioxLSTM [32], DNAGPT [46], and MegaDNA [36], have shifted to causal language modeling. For the scaling of data and models, Omni-DNA is also a decoder-only Transformer trained with causal language modeling on 300 billion nucleotides from RefSeq [33]. Performing an extensive architecture sweep, we further analyze and ablate normalization (LayerNorm vs. RMSNorm) and positional encoding (RoPE vs. ALiBi) to select the most robust configuration.

**Unified multi-task finetuning (Section 3.1).** In the post-training phase, the standard approach is to append a classification head (an MLP) and fine-tune the model independently on each downstream dataset. Using this conventional setup, Omni-DNA achieves state-of-the-art performance on 18 of 26 tasks in both the Nucleotide Transformer [9] and GB benchmarks [13]. We further extend this approach by grouping biologically related tasks and performing joint fine-tuning. The multitask fine-tuning achieves substantial improvements across multiple tasks and obtains the highest overall average accuracy. It demonstrates that Omni-DNA captures transferable biological knowledge during pre-training and exhibits the synergistic effect in the fine-tuning.

Since Omni-DNA is auto-regressive, it can naturally accommodate new tokens as instructions or labels. This enables the use of instruction-following techniques, such as supervised fine-tuning, to enrich the model's vocabulary and enrich the model's vocabulary for more flexible genomic reasoning.

**Effective vocabulary expansion (Section 3.2).** Because most GFMs are pre-trained on unlabeled genomes, their vocabularies are typically restricted to four canonical nucleotides (A, T, C, G) plus a handful of special tokens [9]. In contrast, LLMs routinely operate with vocabularies exceeding $10^4$ sub-word units [25, 41]. This restricts GFM's capacity to both comprehend and generate sequence-level semantics, and conduct the multi-model tasks. To address this, we analyze how to mitigate the distribution shift of vocabulary when adding unseen tokens during post-training. We successfully create a large-scale dataset of text-function mapping, which enables Omni-DNA to directly generate natural functional descriptions of given DNA sequences.

**Scalable long-context reasoning (Section 3.3).** Our final improvement focuses on extending the model's effective context length. Prior approaches such as Caduceus [30], HyenaDNA [24], Evo [22] and MegaDNA [36] replace the standard Transformer with state-space architectures, hierarchical compression, or convolution-based operators to achieve sub-quadratic complexity. However, the whole genome includes long repetitive regions and short meaningful elements (e.g., genes and regulatory regions). These methods do not explicitly exploit such sparsity, limiting their effectiveness at the whole-genome scale. To address this gap, we introduce SEQPACK, a post-training context-extension operator that learns to compress historical tokens into a compact synopsis. Specifically, SeqPack adaptively compresses a DNA sequence of length $N$ into an embedding of length $L$ ($L \ll N$), reducing the complexity from $\mathcal{O}(N^2)$ to $\mathcal{O}(L^2)$. By learning to ignore "junk" regions [27] and retain only informative segments, SeqPack enables efficient ultra-long genomic reasoning while preserving critical positional information.

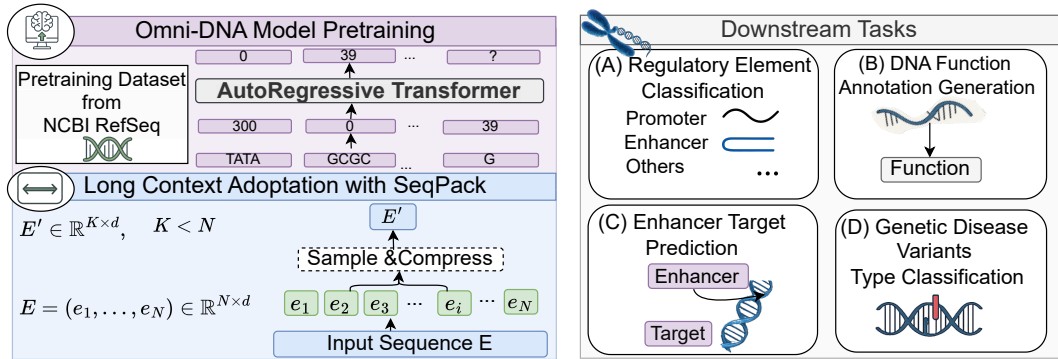

Figure 1: **Overview of the Omni-DNA framework.** Omni-DNA combines large-scale autoregressive pretraining on 300 billion nucleotides with post-training compression via SeqPack to enable efficient long-context DNA modeling. The pretrained model is evaluated separately across diverse downstream tasks, including (A) regulatory element classification, (B) DNA function annotation, (C) enhancer–target interaction prediction, and (D) disease type classification based on genetic variants.

## 2 Revisiting DNA Sequence Pretraining Paradigms

### 2.1 Pretraining Setup

Omni-DNA is pre-trained with causal next-token prediction objective on a 300-billion–nucleotide corpus drawn from NCBI's multi-species assemblies [33]. Data deduplication procedures are described in Appendix C. Following Zhou et al. [48], we adopt Byte-Pair Encoding (BPE) [35] with an initial vocabulary of 4096. Each model processes 250-token contexts ($\approx 1$kbp) in batches of 384 sequences across 8 × A100-40 GB GPUs for 800k steps. The optimization is performed by AdamW ($\beta_1 = 0.9$, $\beta_2 = 0.95$, weight-decay=0.1) with a linear warm-up of 5k steps to a peak learning rate of $3 \times 10^{-4}$, followed by linear decay to $3 \times 10^{-5}$. We also employ mixed precision (bf16 matmuls, fp32 norm stats), global gradient clipping at 1.0, and PyTorch FSDP layer sharding.

### 2.2 Model Architecture

Following Table 1, Omni-DNA adopts a decoder-only Transformer backbone [14], but is *not* a naive transplant of NLP models. We conduct three kinds of ablation studies: normalization layer (bias-free LayerNorm [4] vs. RMSNorm [45]), positional encoding (ALiBi [29] vs. RoPE [40]), and parameter budget. The optimizer, learning-rate scheduler, and tokenizer are kept identical.

Table 1: Architecture of pretrained Omni-DNA models.

| # Params | Layers | Heads | $d_{model}$ | Layer Norm | Pos. Emb. |
|---|---|---|---|---|---|
| 20M | 8 | 8 | 256 | RMSNorm | RoPE |
| 60M | 8 | 8 | 512 | RMSNorm | RoPE |
| 116M | 12 | 12 | 768 | Bias-Free LN | RoPE |
| 116M | 12 | 12 | 768 | RMSNorm | RoPE |
| 116M | 12 | 12 | 768 | Bias-Free LN | ALiBi |
| 116M | 12 | 12 | 768 | RMSNorm | ALiBi |
| 300M | 16 | 16 | 1024 | RMSNorm | RoPE |
| 700M | 16 | 16 | 1536 | RMSNorm | RoPE |
| 1B | 16 | 16 | 2048 | Bias-Free LN | RoPE |

### 2.3 Empirical Insights

> ***Finding 1:*** The combination of RoPE and Bias-Free LayerNorm deliver the best trade-off between compute efficiency and downstream accuracy, whereas ALiBi consistently converges $\sim 15\%$ slower and lags on both promoter and enhancer tasks.

**Explanation.** As shown in Figure 2 (*Left*), all variants exhibit lower training loss with increasing FLOPs, but those using RoPE (green, red) descend more steeply than their ALiBi counterparts (blue, orange). When transferred to the performance of downstream tasks in Figure 2 (*Middle* and *Right*), such as TATA-box recognition and enhancer-activity prediction, RoPE+Bias-Free LayerNorm attains $0.967 \pm 0.002$ and $0.594 \pm 0.006$ accuracy, surpassing all other models. These results suggest that relative phase encoding (RoPE) captures long-range DNA dependencies more effectively than the linear bias of ALiBi, especially when coupled with the unbiased normalizer.

> **Finding 2:** With the same compute larger models tend to obtain lower test loss; however, lower test loss *does not* guarantee higher downstream accuracy

**Explanation.** Figure 3 (*Left*) follows the similar compute–loss scaling law [18] extended from LLMs: with the same compute budget, larger models attain lower test loss (cross-entropy). Yet Figure 3 (*Right*) reveals a decoupling between losses and downstream performance: histone-modification and enhancer tasks saturate once the loss reaches $\approx 4.1$, while promoter and splice tasks remain near-ceiling even for the smallest models (The downstream tasks refer to 18 tasks from Dalla-Torre et al. [9]; finetuning details are in Section 4.1.) This mismatch cautions against relying solely on the loss as a proxy for downstream biological performance.

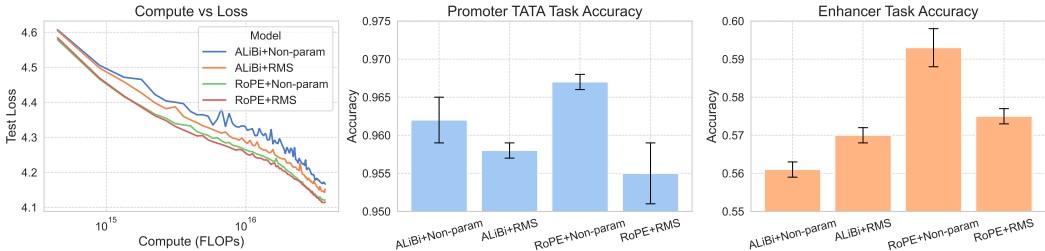

Figure 2: **The ablation of pretraining architectures. Left:** Test loss as a function of cumulative pre-training compute (FLOPs, log–scale) for four 116M-parameter variants that combine ALIBI or ROPE positional biases with bias-free LAYERNORM (*Non-param*) or RMSNORM. **Middle:** Promoter TATA site classification accuracy **Right:** Enhancer activity prediction accuracy. Bars show mean ± std over three fine-tuning runs.

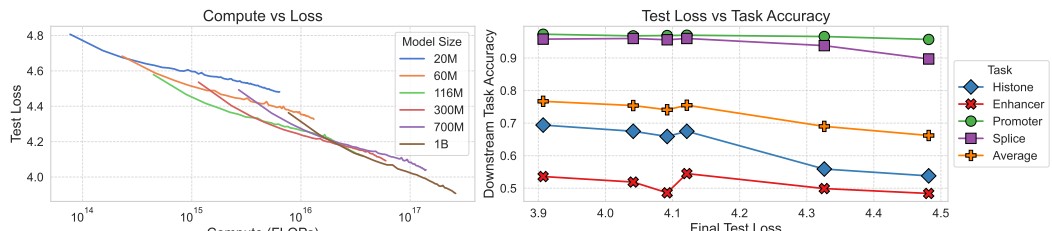

Figure 3: **Compute–Loss Scaling and Its Decoupling from Downstream Accuracy. Left:** Pre-training loss decreases predictably with total compute as model size increases from 20M to 1B parameters, following an approximate power law. **Right:** Final test loss poorly predicts the fine-tuned performance on four representative tasks (Histone, Enhancer, Promoter, Splice).

## 3   Adaptive Post-Training Strategies for GFMs

Here we first explore the new multi-task setting in Section 3.1. Subsequently, we propose novel approaches for vocabulary expansion and long-context compression in Section 3.2 and Section 3.3.

### 3.1   Single-task vs. Multi-task Setting

Current fine-tuning in GFMs for a specific task typically is attaching a linear head (an MLP) to the hidden state of the last token from the final layer, which is then used to predict the class. In the single-task setting, the model is fine-tuned separately for each individual task. In contrast, for the multi-task setting, we first group $k$ related tasks, and jointly fine-tune these $k$ linear heads.

Moreover, since Omni-DNA is auto-regressive, an alternative fine-tuning approach is to introduce new tokens instructions and labels and apply instruction-following techniques, such as supervised fine-tuning (SFT). For multiple tasks, we add task-specific prompt tokens and class label tokens, enabling the model to learn $k$ tasks simultaneously.

## 3.2 Vocabulary Expansion

To enable the model to handle prompts, answer tokens, or label words that are *absent* from the pre-training alphabet during SFT, the original vocabulary $V_o$ must be expanded to $V = V_o \cup V_z$, where $V_z$ represents the newly introduced symbols (e.g., task tags, response delimiters, class labels).

**Vocabulary distribution shift problem.** Let $h_{t-1} \in \mathbb{R}^d$ be the decoder state at step $t-1$ and $E = [\,e_v\,]_{v \in V} \in \mathbb{R}^{d \times |V|}$ the output-embedding matrix whose *columns* are the token vectors $e_v$. With the expanded vocabulary, the softmax becomes

$$p_\theta(x_t \mid x_{1:t-1}) = \frac{\exp(h_{t-1}^\top e_{x_t})}{\underbrace{\sum_{m \in V_o} \exp(h_{t-1}^\top e_m)}_{Z} + \underbrace{\sum_{n \in V_z} \exp(h_{t-1}^\top e_n)}_{Z'}}. \tag{1}$$

For a pre-existing token $x_t \in V_o$, its probability under the original softmax would have been $p_\theta^{\text{old}}(x_t \mid x_{1:t-1}) = \exp(h_{t-1}^\top e_{x_t})/Z$. Comparing with Equation 1 yields the shrinkage factor:

$$p_\theta(x_t \mid x_{1:t-1}) = p_\theta^{\text{old}}(x_t \mid x_{1:t-1}) \times \frac{Z}{Z + Z'} = p_\theta^{\text{old}}(x_t \mid x_{1:t-1}) \times \frac{1}{1 + Z'/Z}. \tag{2}$$

Thus every *pre-existing* token probability is decayed by $1/(1 + Z'/Z)$—a shift that can slow convergence and, if unaddressed, cause catastrophic forgetting.

To mitigate distribution shift, we integrate two techniques: **(1) NEFTune** [16]: isotropic Gaussian noise is added to all token embeddings during SFT, acting as dropout in representation space and discouraging the model from over-relying on a few high-frequency additions. **(2) Key-token replication**: for classification tasks, we duplicate each label token $\alpha$ times (we use $\alpha = 5$) within the target string. This boosts its relative frequency, ensuring that gradients from the label position dominate those from task prompts. In practice, we also keep $|V_z|$ to the minimum required.

## 3.3 SEQPACK: Long-Context Sequence Compression

Let $x_{1:N} = (x_1, \ldots, x_N)$ denote a DNA sequence of length $N \gg L_{\text{nat}}$, where $L_{\text{nat}}$ is the *native context length* of a pretrained backbone $f_\theta$ (typically 1–16k tokens). We tokenize DNA sequences into $k$-mers and apply the embedding table $\Phi : \mathcal{V} \to \mathbb{R}^d$ to obtain the embedding matrix $E = \Phi(x_{1:N}) \in \mathbb{R}^{N \times d}$. The objective of SEQPACK is to learn a differentiable compression operator

$$\mathcal{C}_L : \mathbb{R}^{N \times d} \longrightarrow \mathbb{R}^{K \times d}, K \leq L_{\text{nat}} \tag{3}$$

which can identify and preserve the most informative $K$ tokens based on their positions and content before the computation-heavy layers.

### 3.3.1 Position-Aware Learnable Sampling

To realize the compression operator $\mathcal{C}_L$, we estimate the token importance by a trainable *temperature vector* $\theta \in \mathbb{R}^N$. In detail, the importance weight of each position $i$ is defined by

$$\pi_i = \frac{\exp(\theta_i)}{\sum_{j=1}^N \exp(\theta_j)}. \tag{4}$$

Because hard sampling is non-differentiable, we employ the Gumbel-Softmax trick [15]:

$$\text{logit}_i = \log \pi_i + g_i, \qquad g_i \sim \text{Gumbel}(0, 1), \tag{5}$$

followed by a top-$K$ operator that keeps the $K$ highest-scoring indices. Concatenating the selected rows of $E$ with a fixed suffix of the most recent $K$ tokens yields the compressed representation $E' = \mathcal{C}_L(E)$, guaranteeing that the downstream backbone receives at most $K$ tokens.

### 3.3.2 Geometric Splitting

Applying a global top-$K$ over $N$ tokens costs $O(N \log K)$ time, which is untenable for ultra-long sequences. To avoid this bottleneck, we introduce a *geometric splitting* scheduler that partitions the sequence into exponentially growing segments and performs local sampling within each segment.

---

**Algorithm 1** SEQPACK compression layer (inference mode)

---

**Require:** Embeddings $E \in \mathbb{R}^{N \times d}$, the number of preserved tail tokens $K_{\text{tail}}$, the total preserved tokens $K$, the learned temperature vector $\theta$
**Ensure:** Compressed sequence $E' \in \mathbb{R}^{K \times d}$
 1: $H \leftarrow N - K_{\text{tail}} - 1$          ▷ history length (excluding $e_1$ and tailed tokens)
 2: $B \leftarrow K - K_{\text{tail}} - 1$          ▷ history budget
 3: **if** $H \leq B$ **then**
 4:      **return** $E$          ▷ no compression needed
 5: **end if**
 6: **Segment** $e_{2:N-K_{\text{tail}}}$ into geometric blocks $\{S_i\}_{i=1}^{M}$
 7: **for all** $S_i$ **do**
 8:      Compute the sampling budget $b_i$ via Equation 7
 9:      Compute the importance weight $\pi_j^i$ via Equation 8
10:      Sample to obtain compressed output $E_i'$ based on $b_i$ and $\pi_j^i$
11: **end for**
12: $E' \leftarrow [\, e_1 \,;\, E_1' \,;\, \ldots \,;\, E_M' \,;\, e_{N-K_{\text{tail}}:N} \,]$
13: **return** $E'$

---

Given input embeddings $E \in \mathbb{R}^{N \times d}$ and compression target $K$, we divide the sequence into $M = \lceil \log_2(N) \rceil$ segments with exponentially increasing sizes. The starting point of these segments $i \in [1, M-1]$ is

$$p_{i+1} = p_i + \min(2^{i-1}, N - p_i) \tag{6}$$

where the boundary conditions are $p_1 = 1$ and $p_{M+1} = N + 1$. For each segment $S_i = E_{[p_i:p_{i+1}-1]}$, we allocate a sampling budget based on the length:

$$b_i = \max\left(1, |S_i| \cdot \frac{K}{N}\right). \tag{7}$$

We then sample $b_i$ tokens from each segment according to the segment-normalized importance weights following Equation 4:

$$\pi_j^i = \frac{\exp(\theta_j)}{\sum_{k=p_i}^{p_{i+1}-1} \exp(\theta_k)} \qquad j \in [p_i, p_{i+1} - 1], \tag{8}$$

From each segment, we obtain sampled $E_i'$ and the final compressed output is $E' = [E_1'; \ldots; E_M']$. To further analyze the efficiency of geometric splitting, we present Proposition 1 with the proof in Appendix B, which establishes the linear-time complexity of the sampling process.

**Proposition 1** (Linear-time compression under geometric splitting). *Let $E \in \mathbb{R}^{N \times d}$ be token embeddings and let $K \leq L_{nat}$ be the target context length. Using the geometric splitting scheduler defined above, the expected time to obtain the compressed sequence $E' \in \mathbb{R}^{K \times d}$ is $\Theta(N)$.*

### 3.3.3 Compression with Positional Bias Preservation

We implement SEQPACK as a lightweight PyTorch layer that is inserted *once* before the first attention block during post-training. To preserve positional inductive biases, we exclude the first token and the final $K_{\text{tail}}$ tokens from compression. This design is motivated by the attention sink phenomenon [42], where attention to the first token can dominate attention scores. In DNA-pretrained transformers, we observe similar sink effects at both the beginning and end of the sequence (Appendix G). The overall procedure is described in Algorithm 1.

## 4 Experiments

We benchmark Omni-DNA integrated with SEQPACK against leading genomic foundation models across four types of representative tasks. (i) Gene regulatory element classification: identify functional regions that control gene expression; (ii) Pathogenic variant effect prediction: assess the functional impact of genetic mutations linked to disease; (iii) Enhancer–target gene interaction mapping: resolve the complex spatial relationships between regulatory elements and their target genes; and (iv) Sequence-conditioned functional interpretation generation: a novel task introduced in this work, supported by a carefully curated dataset derived from high-quality NCBI genome assemblies.

Table 2: **Comparison of Omni-DNA with existing GFMs on 18 NT downstream tasks.** The family of Omni-DNA obtains the best average result. In splice tasks, Omni-DNA performs lower than NT models but still surpasses other models.

| Metric | CADUCEUS-PH (1.9 M) | CADUCEUS-PS (1.9 M) | DNABERT2 (117 M) | NTv2 (500 M) | NT2.5B (2.5 B) | HyenaDNA (1.6 M) | Omni-DNA (116 M) | Omni-DNA (1 B) |
|---|---|---|---|---|---|---|---|---|
| H3 | $0.815 \pm 0.048$ | $0.799 \pm 0.029$ | $0.785 \pm 0.033$ | $0.784 \pm 0.047$ | 0.814 | $0.779 \pm 0.037$ | $\underline{0.818} \pm 0.005$ | $\mathbf{0.824} \pm 0.032$ |
| H3K14AC | $0.631 \pm 0.026$ | $0.541 \pm 0.212$ | $0.516 \pm 0.028$ | $0.551 \pm 0.021$ | 0.550 | $0.612 \pm 0.065$ | $\underline{0.685} \pm 0.014$ | $\mathbf{0.697} \pm 0.077$ |
| H3K36ME3 | $0.601 \pm 0.129$ | $0.609 \pm 0.109$ | $0.591 \pm 0.020$ | $0.625 \pm 0.013$ | 0.632 | $0.613 \pm 0.041$ | $\underline{0.661} \pm 0.013$ | $\mathbf{0.686} \pm 0.002$ |
| H3K4ME1 | $0.523 \pm 0.039$ | $0.488 \pm 0.102$ | $0.511 \pm 0.028$ | $0.550 \pm 0.021$ | 0.559 | $0.512 \pm 0.024$ | $\underline{0.577} \pm 0.083$ | $\mathbf{0.617} \pm 0.000$ |
| H3K4ME2 | $0.487 \pm 0.170$ | $0.388 \pm 0.101$ | $0.336 \pm 0.040$ | $0.319 \pm 0.045$ | 0.326 | $0.455 \pm 0.095$ | $\mathbf{0.576} \pm 0.003$ | $\underline{0.547} \pm 0.006$ |
| H3K4ME3 | $0.544 \pm 0.045$ | $0.440 \pm 0.202$ | $0.352 \pm 0.077$ | $0.410 \pm 0.033$ | 0.421 | $0.549 \pm 0.056$ | $\underline{0.587} \pm 0.222$ | $\mathbf{0.642} \pm 0.001$ |
| H3K79ME3 | $0.697 \pm 0.077$ | $0.676 \pm 0.026$ | $0.613 \pm 0.030$ | $0.626 \pm 0.026$ | 0.642 | $0.672 \pm 0.048$ | $\underline{0.718} \pm 0.027$ | $\mathbf{0.752} \pm 0.007$ |
| H3K9AC | $0.622 \pm 0.030$ | $0.604 \pm 0.048$ | $0.542 \pm 0.029$ | $0.562 \pm 0.040$ | 0.575 | $0.581 \pm 0.061$ | $\underline{0.658} \pm 0.029$ | $\mathbf{0.701} \pm 0.002$ |
| H4 | $0.811 \pm 0.022$ | $0.789 \pm 0.020$ | $0.796 \pm 0.027$ | $0.799 \pm 0.025$ | $\underline{0.822}$ | $0.763 \pm 0.044$ | $0.802 \pm 0.002$ | $\mathbf{0.822} \pm 0.005$ |
| H4AC | $0.621 \pm 0.054$ | $0.525 \pm 0.240$ | $0.463 \pm 0.041$ | $0.495 \pm 0.032$ | 0.501 | $0.564 \pm 0.038$ | $\mathbf{0.663} \pm 0.029$ | $\underline{0.652} \pm 0.001$ |
| Enhancer | $0.546 \pm 0.073$ | $0.491 \pm 0.066$ | $0.516 \pm 0.098$ | $0.548 \pm 0.144$ | 0.580 | $0.517 \pm 0.117$ | $\mathbf{0.593} \pm 0.005$ | $0.580 \pm 0.018$ |
| Enhancer Types | $0.439 \pm 0.054$ | $0.416 \pm 0.095$ | $0.423 \pm 0.051$ | $0.424 \pm 0.132$ | 0.474 | $0.386 \pm 0.185$ | $\underline{0.498} \pm 0.001$ | $\mathbf{0.492} \pm 0.023$ |
| Promoter: ALL | $0.970 \pm 0.004$ | $0.967 \pm 0.004$ | $0.971 \pm 0.006$ | $\underline{0.976} \pm 0.006$ | **0.974** | $0.960 \pm 0.005$ | $0.973 \pm 0.001$ | $0.973 \pm 0.001$ |
| Promoter: NONTATA | $0.969 \pm 0.011$ | $0.968 \pm 0.006$ | $0.972 \pm 0.005$ | $\underline{0.976} \pm 0.005$ | **0.977** | $0.959 \pm 0.008$ | $0.972 \pm 0.016$ | $0.975 \pm 0.001$ |
| Promoter: TATA | $0.953 \pm 0.016$ | $0.957 \pm 0.015$ | $0.955 \pm 0.021$ | $0.966 \pm 0.013$ | 0.964 | $0.944 \pm 0.040$ | $\underline{0.967} \pm 0.001$ | $\mathbf{0.973} \pm 0.002$ |
| Splice All | $0.940 \pm 0.027$ | $0.927 \pm 0.021$ | $0.939 \pm 0.009$ | $\underline{0.983} \pm 0.008$ | **0.983** | $0.956 \pm 0.011$ | $0.927 \pm 0.025$ | $0.941 \pm 0.003$ |
| Splice Acceptor | $0.937 \pm 0.033$ | $0.936 \pm 0.077$ | $0.975 \pm 0.006$ | $\underline{0.981} \pm 0.011$ | **0.990** | $0.958 \pm 0.010$ | $0.968 \pm 0.002$ | $0.972 \pm 0.001$ |
| Splice Donor | $0.948 \pm 0.025$ | $0.874 \pm 0.289$ | $0.963 \pm 0.006$ | $\mathbf{0.985} \pm 0.022$ | $\underline{0.984}$ | $0.949 \pm 0.024$ | $0.951 \pm 0.007$ | $0.963 \pm 0.001$ |
| **Average** | 0.715 | 0.689 | 0.679 | 0.698 | 0.709 | 0.707 | $\underline{0.755}$ | **0.767** |

Table 3: **Performance of Omni-DNA across 8 tasks in the genomic benchmark.** Omni-DNA (116M) achieves the highest average.

| Task Name | CNN (264K) | HYENADNA (436K) | CADUCEUS -PH (470K) | DNABERT2 (117M) | Omni-DNA (116M) |
|---|---|---|---|---|---|
| MOUSE ENHANCERS | $0.715 \pm 0.087$ | $0.780 \pm 0.025$ | $0.754 \pm 0.074$ | $\underline{0.792} \pm 0.031$ | $\mathbf{0.799} \pm 0.004$ |
| CODING VS. INTERGENOMIC | $0.892 \pm 0.008$ | $0.904 \pm 0.005$ | $0.915 \pm 0.003$ | $\mathbf{0.949} \pm 0.002$ | $\underline{0.942} \pm 0.010$ |
| HUMAN VS. WORM | $0.942 \pm 0.002$ | $0.964 \pm 0.002$ | $0.973 \pm 0.001$ | $0.975 \pm 0.002$ | $\mathbf{0.976} \pm 0.001$ |
| HUMAN ENHANCERS COHN | $0.702 \pm 0.021$ | $0.729 \pm 0.014$ | $\mathbf{0.747} \pm 0.004$ | $0.714 \pm 0.025$ | $\underline{0.738} \pm 0.002$ |
| HUMAN ENHANCER ENSEMBL | $0.744 \pm 0.122$ | $0.849 \pm 0.006$ | $\underline{0.893} \pm 0.008$ | $0.891 \pm 0.051$ | $\mathbf{0.919} \pm 0.021$ |
| HUMAN REGULATORY | $0.872 \pm 0.005$ | $0.869 \pm 0.012$ | $\underline{0.872} \pm 0.011$ | $0.852 \pm 0.024$ | $\mathbf{0.895} \pm 0.012$ |
| HUMAN OCR ENSEMBL | $0.698 \pm 0.013$ | $0.783 \pm 0.007$ | $\mathbf{0.828} \pm 0.006$ | $0.789 \pm 0.012$ | $\underline{0.791} \pm 0.001$ |
| HUMAN NONTATA PROMOTERS | $0.861 \pm 0.009$ | $0.944 \pm 0.002$ | $0.946 \pm 0.007$ | $0.912 \pm 0.013$ | $\mathbf{0.968} \pm 0.013$ |
| **Average** | 0.803 | 0.853 | 0.866 | 0.859 | **0.879** |

## 4.1 Gene Regulatory Element Classification

**Nucleotide Transformer Downstream** Nucleotide Transformer Downstream [9] includes 18 tasks both binary and multi-class classification tasks for epigenetic marker [28], promoter [26], enhancer and splice site prediction. Closely following the evaluation metrics from Caduceus [31], Matthews Correlation Coefficient (MCC) is used for histone marker and enhancer, while F1-score is used for splice and promoter classification. To compare Omni-DNA against Nucleotide Transformer V2(NTv2) [9], DNABERT2 [48], HyenaDNA [23], Caduceus [31]. We perform 10-fold cross-validation to select the hyperparameter for each model as detailed in Appendix D, and the detailed task-specific results for NT-Downstream tasks in Table 2. Omni-DNA (1B) and Omni-DNA (116M) achieve the best average performance of 0.767 and the second of 0.755, and they achieve superior performance on 13 out of 18 benchmark tasks, surpassing competing methods. For the remaining five tasks, Omni-DNA (1B) ranks second and third in Promoter:ALL and Promoter:NonTATA. In the three splice site classification tasks, Omni-DNA models perform lower than NT models but still surpass other models.

**Genomic Benchmark** Genomic Benchmark (GB) [13] contains eight DNA regulatory element classification tasks, similar to NT downstream tasks, with additional tasks on species classification and gene regulatory element classification on mouse. We compare model performance of Omni-DNA (116M) with DNABERT2, CADUCEUS-PH, and HynnaDNA, and show the results in Table 3. Omni-DNA (116M) is ranked first in five out of eight tasks and second in the remaining three, while achieving the highest average score. Compared to DNABERT-2, which has a similar model size, Omni-DNA (116M) outperformed DNABERT-2 in seven out of eight tasks.

**Multi-tasking.** We extend pretrained Omni-DNA (116M) and NTv2 (500M) models to a multi-task setting and compare their performance with their single tasks counter-parts. In particular, 10 related tasks from NT Downstream [9]—H3, H3K14AC, H3K36ME3, H3K4ME1, H3K4ME2, H3K4ME3, H3K79ME3, H3K9AC, H4, and H4AC—are considered. These tasks are interconnected due to the biological correlation between acetylation and methylation effects [28]. The objective here is to train a model capable of addressing all 10 tasks with a single fine-tuning.

Two approaches are compared: **Multi-head fine-tuning** attaches task-specific linear heads to a shared Transformer backbone. **Next-Token prediction** augments the tokenizer with three label tokens $\{0, 1, 2\}$ and ten task identifiers $\{\texttt{task}_1 \ldots \texttt{task}_{10}\}$; the model autoregressively emits the label token, casting classification as language modelling.

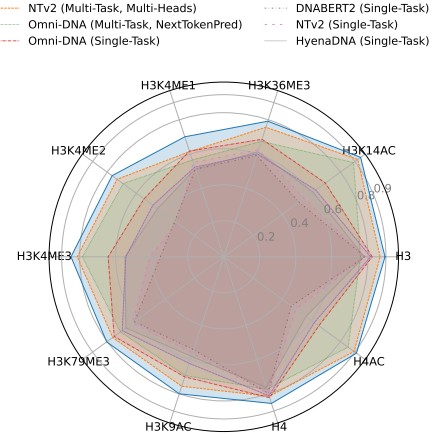

Figure 4: **Multi-task fine-tuning boosts accuracy.** Classification MCC on ten nucleosome-target (NT) tasks. Multi-head outperform NTP multi-tasking, but both surpass single-task baselines.

Figure 4 shows *Omni-DNA (multitask,multi-head)* and *NTv2 (multitask,multi-head)* strictly outperform their single-task counterparts on all tasks, while *Omni-DNA (multi-task,NTP)* also achieves higher accuracy in half of the tasks. We hereby summarize the results as following finding.

> **Finding 3:** Jointly fine-tuning on biologically related tasks consistently outperforms independent single-task fine-tuning.

## 4.2 Enhancer–Target Interaction Prediction

**Task** Enhancers are distal regulatory elements that can reside *kilobases* to *megabases* from the genes they activate, in contrast to promoters that typically lie immediately upstream of a transcription start site. We adopt the curated benchmark from Cheng et al. [7] originated from [11, 12, 34], which contains 2,602 experimentally validated enhancer–target pairs plus matched negative controls. Each sample is a 450 kb genomic window, and the data are split 8:1:1 into training, validation, and test sets. The task is binary classification: does the window contain an *effective* enhancer–target interaction? *Because accurate prediction requires modelling dependencies across hundreds of kilobases, this task is intrinsically more challenging than the shorter-context tasks assessed in the previous section.*

**Models** Because 450kb exceeds the context limits of NTv2 and DNABERT2, both models ran out of memory (OOM) and are therefore omitted. For Omni-DNA (116M) we raise the *native context length* to 2048 tokens to accommodate longer sequences. Additional long-range baselines include HYENADNA (medium-450K) and CADUCEUS (PH/PS). We also report an expert "upper-bound" model, Activity-By-Contact (ABC), which leverages DNase-seq, ChIP-seq, and Hi-C matrices, and a lightweight CNN baseline from Cheng et al. [7]. The performance is measured by AUROC.

Table 4: **Enhancer–Target Sequence Classification**

**Results** For HYENADNA, CADUCEUS, and Omni-DNA, we append a single linear classification head and mean-pool the final hidden states over the sequence length. Models are fine-tuned for at most 20 epochs; learning rate and batch size are selected via a small grid search (see Table 8). As summarized in Table 4, Omni-DNA attains an AUROC of **0.895**, outperforming all neural baselines and closing much of the gap to the assay-rich ABC upper bound. In Appendix H, we provide additional ablation studies on the computational overhead and compression ratio of SeqPack.

| Model | AUROC |
|---|---|
| ABC (expert) | 0.926 |
| CNN | 0.797 |
| HyenaDNA | 0.828 |
| CADUCEUS-PH | 0.826 |
| CADUCEUS-PS | 0.821 |
| Omni-DNA | **0.895** |

Table 5: **Context-length sensitivity (*left*) and variant-type classification (*right*).** *Left*: F1 score of Omni-DNA on LC as a function of input length. *Right*: F1 score (↑) on LC and ICCs.

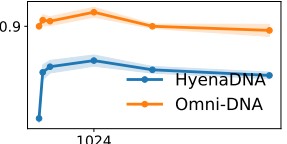

|       | HyenaDNA      | DNABERT2       | NTv2          | Omni-DNA        |
|-------|---------------|----------------|---------------|-----------------|
| **LC**   | 0.83±0.02 | 0.71± 0.03 | 0.75± 0.04 | **0.93±0.01** |
| **ICCs** | 0.78±0.08 | 0.49±0.02  | 0.75±0.00  | **0.89±0.01** |

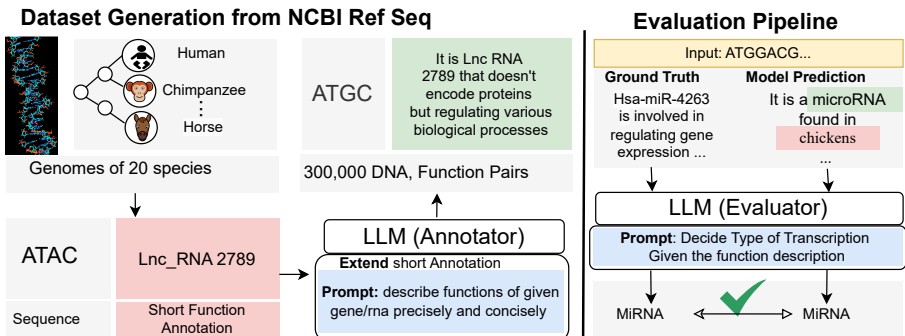

Figure 5: **SEQ2FUNC construction and evaluation pipeline.** Genomic windows extracted from 20 NCBI species are clustered by transcriptional evidence and passed to a LLM that produces concise, expert-style function descriptions. After agreement filtering and manual vetting the process yields 300k sequence–function pairs covering 10 functional classes.

## 4.3 Genetic Disease Variant–Type Classification

Moving beyond binary pathogenicity prediction [21], we address the more challenging task of directly inferring the disease category from a single–nucleotide variant (SNV), encoded as the triple (reference sequence, mutated sequence, label). We consider two clinically salient groups: **(i) Lung Cancer (LC)** with 11 disease types, and **(ii) Inherited Cardiac Conditions (ICCs)** with 1108 disease types.

Starting from the full ClinVar release (14810 unique diseases and ∼3 M variants), variants annotated with any of the **LC** or **ICCs** labels are marked positive; negatives are drawn at random from the remaining diseases. The resulting training sets contain 7,876 LC variants and 532,621 ICC variants. Each example is therefore a triple $(r, m, y)$, where $r$ is the reference sequence, $m$ the mutated sequence, and $y \in \{0, 1\}$ denotes the absence or presence of the target disease class.

The left of Table 5 shows that extending the context length improves performance up to 1024bp, with negligible gains thereafter. We fine-tune Omni-DNA (116M), DNABERT2, NTv2 (500M), and HyenaDNA-medium (450K) based on the procedure in Appendix E. The F1 scores are summarized in the right of Table 5, where Omni-DNA delivers the strongest performance on both disease groups.

## 4.4 Sequence-to-Function Generation

**Dataset.** Publicly available functional labels are sparse, making automatic interpretation of raw DNA sequences challenging. To bridge this gap, we release **SEQ2FUNC**, a corpus of 300k high-quality *(sequence, function)* pairs (Figure 5). Starting from coarse annotations in the NCBI multi-species assemblies [6], we prompt a state-of-the-art LLM to generate refined, natural-language descriptions. Each entry is validated through agreement between two independent LLM passes, with low-confidence cases discarded. The final dataset spans 20 species and 10 broad functional categories, with detailed statistics provided in Appendix F. We follow an 80/10/10 train/validation/test split.

**Task and training protocol.** Given a DNA sequence, the objective is to generate a textual description for its functional annotation. We first extend the Omni-DNA vocabulary from 4096 DNA tokens to 8173 tokens (adding 4077 wordpieces) and then fine-tune with next-token prediction on

> **Model Input:** What is the function of this sequence?GGCTG...TTTTCTGA
>
> **Ground Truth:** Olfactory receptor 5A2-like (OR5A2L) is a member of the olfactory receptor gene family, which is involved in the detection of volatile substances , contributing to smell sense.
>
> **Omni-DNA**: The mRNA for olfactory receptor family 5 subfamily B member 109 ( OR5B109 ) encodes a protein that is part of the olfactory receptor family. These receptors are involved in the detection of odorant molecules and and play a crucial role in the sense of smell.

Figure 6: **Demonstration of Omni-DNA's functional description generation.** Given a DNA sequence, Omni-DNA could generate a natural language description for functional annotations.

the **SEQ2FUNC** training set, consisting of concatenated [DNA]‖[description] sequences. We use AdamW and NEFTune to train for up to 10 epochs with early stopping on validation perplexity. An LLM-based evaluator maps each generated description to one of 10 functional classes, allowing quantitative comparison with ground truth via weighted F1 score and Matthews correlation coefficient (MCC). Since existing genomic models cannot produce free-form text, we benchmark against GPT-4o in zero-shot mode (used prompts are shown in Table 9), with a random guesser as a lower bound.

**Results.** The qualitative examples in Figure 6 and Appendix F show that Omni-DNA@FT produces fluent and biologically plausible explanations and correctly identifies an unseen olfactory-receptor locus. Moreover, the quantitative results in Table 6 show that Omni-DNA outperforms GPT-4o in zero-shot generation by $7.1\%$ in weighted F1 and $38.2\%$ in MCC, substantially exceeding the random baseline.

Table 6: Performance on the SEQ2FUNC test set.

| Model | Weighted F1 | MCC |
|---|---|---|
| Omni-DNA @FT | **0.730** | **0.367** |
| GPT-4o (zero-shot) | 0.659 | $-0.015$ |
| Random Guess | 0.483 | 0.008 |

## 5 Conclusion

In this paper, we introduced Omni-DNA, a framework for exploring optimal strategies in pre-training and task adaptation for genomic modeling. Empirical analysis identified that the combination of non-parametric bias-free layer normalization with RoPE positional embeddings delivers the best trade-off between computational efficiency and downstream accuracy. We further demonstrate that multi-task fine-tuning on biologically related tasks substantially enhances performance. To extend the applicability of Omni-DNA, we develop SEQPACK, enabling efficient processing of ultra-long DNA sequences, and presentSEQ2FUNC, which empowers Omni-DNA to annotate DNA sequence functions using natural language. Across a range of downstream tasks, including regulatory element classification, enhancer–target interaction prediction, genetic disease variant classification, and DNA function annotation generation, Omni-DNA achieves state-of-the-art performance. We envision Omni-DNA as a step for scaling and generalizing genomic language models toward broader biological tasks.

## Acknowledgments and Disclosure of Funding

Zehui Li is grateful to his family—Jinghai Li, Huiqing Wang, and Yiqiu Sun for their support.

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

# A  Limitations

Despite these advances, limitations exists. Naive approach to explore the architecture choices through pretraining could be costly. Future research could streamline architecture optimization using automated techniques such as Zero-Shot NAS [20]. Additionally, integrating reinforcement learning for post-training methods could enhance efficiency and address the generalizability issue with supervised fune-tuning [8], although effective reward design remains challenging [37, 43].

# B  Proofs of Propositions

*Proof.* We bound the cost of constructing the compressed sequence $E'$ under the geometric splitting scheduler.

(a) **Segment construction.** The sequence is partitioned into $M = \lceil \log_2 N \rceil$ segments whose lengths grow geometrically as $|S_i| \leq 2^{i-1}$. Forming the segment boundaries therefore requires $O(M) = O(\log N)$ operations.

(b) **Local sampling.** Let $b_i$ $(1 \leq i \leq M)$ be the sampling budget of segment $S_i$. Selecting $b_i$ indices with a heap-based top-$b_i$ (or an equivalent QuickSelect routine) costs $O(|S_i| \log b_i)$ time. Because the scheduler guarantees $b_i \leq |S_i|$ and, in practice, $b_i \ll |S_i|$ for long histories, we may upper-bound the per-segment cost by $O(|S_i| \log |S_i|)$.

(c) **Total cost.** Summing over all segments,

$$\sum_{i=1}^{M} O\big(|S_i| \log |S_i|\big) = \sum_{i=1}^{M} O\big(2^{i-1}(i-1)\big).$$

Since $\sum_{i=1}^{M} 2^{i-1} = N$ and $(i-1) \leq \log_2 |S_i|$, the series is dominated by the linear term:

$$\sum_{i=1}^{M} 2^{i-1}(i-1) = O(N).$$

Hence the overall expected running time is $O(N) + O(\log N) = \Theta(N)$, in contrast to the $O(N \log K)$ required by a global top-$K$ selection. $\square$

Table 7: Comparison of pretraining setup between Omni-DNA, DNABERT2, and NT-transformer.

| | Omni-DNA (six sizes) | DNABERT2 (116M) | NT-transformer (5 sizes) |
|---|---|---|---|
| Layer norm type | non-parametric/RMSNorm | parametric | parametric |
| Positional embeddings | RoPE | ALiBi | RoPE |
| Attention variant | full | full | full |
| Biases | none | in both LN and Attention | in both LN and Attention |
| Block type | sequential | sequential | sequential |
| Activation | SwiGLU | GEGLU | SwiGLU |
| Sequence length (In Tokens) | 250 | 128 | 1000 |
| Batch size (Across GPUs) | 384 | 4096 | 512 |
| Total Steps | 800000 | 150000 | 16, 000 |
| Warmup steps | 5000 | - | 16,000 |
| Peak LR | 3.0E-04 | 5.0E-04 | 1.0E-04 |
| Minimum LR | 3.0E-05 | 0 | 5.0E-05 |
| Weight decay | 0.1 | 1e-5 | 0.1 |
| Beta1 | 0.9 | 0.9 | 0.9 |
| Beta2 | 0.95 | 0.98 | 0.999 |
| Epsilon | 1.0E-05 | 1.0E-05 | 1.0E-08 |
| LR schedule | linear | linear | linear |
| Gradient clipping | global 1.0 | - | - |
| Training Precision | Mxied (bf16 for most operations) | - | - |
| Training Framework | Pytorch Built-in FSDP | Pytorch HuggingFace trainer | - |
| Training Duration (min) | 8 GPUs* 1 days | 8GPUs * 14 days | 128 GPUs*1days |
| Training Duration (max) | 8 GPUs* 7 days | 8GPUs * 14 days | 128 GPUs*28days |
| GPU Types | A100 Nvidia GPU 40GB | Nvidia RTX 2080Ti GPUs. | A100 Nvidia GPU |

## C  Detailed Deduplication Process

### C.1  Handling Non-{A, T, G, C} Characters

The raw genome data from the reference genome includes additional characters beyond {A, T, G, C}, which represent ambiguities or gaps. These characters are defined as follows:

- **N**: Represents any nucleotide (A, T, G, or C).
- **R**: Represents purines (A or G).
- **Y**: Represents pyrimidines (C or T).
- Other characters (e.g., **W**, **S**, **K**, **M**, etc.) represent specific nucleotide subsets or unknown bases.

These characters were removed during preprocessing to retain only the core nucleotide sequences ({A, T, G, C}), ensuring consistency and facilitating the deduplication process.

### C.2  Chunk-Based Deduplication

The deduplication procedure was performed as follows:

1. **Chunking**: The genome was divided into non-overlapping chunks of 1024 base pairs.
2. **Exact Matching**: Identical sequences across the dataset were identified and removed.
3. **Efficiency**: This step utilized hashing techniques and optimized string comparison algorithms to handle the large dataset efficiently.

## D  Finetuning with Classification Head

**Basic Setup**   We closely follow the setup from Caduceus [31] for evaluating Omni-DNA. The following models are compared: Omni-DNA, DNABERT-2 [48], NT-Transformer [9], HyenaDNA [23], and the Caduceus models [31].

For NT Downstream tasks, we use a maximum fine-tuning epoch of 20, while for the Genomic Benchmark (GB) tasks, we use a maximum of 10 epochs. Both NT Downstream and GB tasks include a training set and a test set. For hyperparameter search, 10% of the training set is reserved as a validation set.

**Hardware & Framework**   All fine-tuning is conducted on a single NVIDIA A100 40GB GPU. We utilize the Hugging Face `Trainer` API for full-size fine-tuning of the pretrained checkpoints. Models are loaded using the `AutoModelForSequenceClassification` class, which automatically adds a linear layer on top for sequence classification.

Our experimental results align with those reported in Caduceus [31]. For consistency, we report statistics from [31] for two Caduceus models, DNABERT-2, NT-Transformer 500M, and HyenaDNA in NT Downstream tasks, as well as CNN, HyenaDNA, and CADUCEUS-PH in GB tasks. For NT2.5B, we use the results from its original paper [9].

We replicate experiments for the following models: NT50M, NT100M, NT250M, along with six Omni-DNA models on 18 NT Downstream tasks. Additionally, we fine-tune DNABERT-2 and Omni-DNA 116M on the eight Genomic Benchmark tasks.

**Hyperparameters**   10-fold cross validation is performed for each model to decide the best hyperparameters provided in table 8.

**Additional Notes**   We observe a performance gap in HyenaDNA between our replication results, which align with those of [31], and the results reported in the original paper [23], both on NT downstream and GB tasks. Our replication follows our own setup, yielding results consistent with [31]. Upon reviewing the code within the container image provided by [23], we identified additional techniques such as Exponential Moving Average (EMA) and data augmentation that were employed, which may account for the discrepancy. In this work, we use HyenaDNA results obtained without these techniques as the baseline.

Table 8: Finetuning hyperparameters.

| Hyperparameter | Value |
|---|---|
| Peak learning rate | $[3 \times 10^{-4}, 1 \times 10^{-5}, 5 \times 10^{-6}]$ |
| Per device train batch size | [8,16,32] |
| Max gradient norm | 1.0 |
| Weight decay | 0.1 |
| Adam $\beta_1$ | 0.9 |
| Adam $\beta_2$ | 0.999 |
| Adam $\epsilon$ | $1 \times 10^{-8}$ |
| Optimizer | ADAMW |

# E  Genetic Disease Variant Type Classification

**Finetuning procedure.**  For every backbone (NTv2, Omni-DNA, DNABERT2, HyenaDNA) we wrap the pretrained model with a lightweight classification head that follows the implementation in algorithm 2. Each sample is encoded as a pair of sequences—reference (`ref`) and alternate (`alt`). Both sequences are passed independently through the frozen backbone to obtain the last–layer hidden states.

- **Encoder backbones**: we take the first-token representation (`[CLS]`).
- **Decoder backbones**: we take the representation of the last non-padding token in each sequence.

The two vectors are optionally transformed by a learnable pooler (a linear layer followed by $\tanh$). We then compute their difference, $\Delta h = h_{\text{alt}} - h_{\text{ref}}$, and feed it to a single fully connected layer that produces class logits. Cross-entropy loss is minimised for at most 100 epochs; a coarse grid search (summarised in table 8) is used to select learning rate, batch size, and weight decay.

# F  Functional Annotation Generation

**Dataset Construction Process**  The Seq2Func dataset is constructed using genomic sequences sourced from the **NCBI Reference Sequence (RefSeq)** database, incorporating sequences from **20 different species**. The selected species and their genome assemblies include:

- GCF-000001405.40, human
- GCF-000001635.27, mouse
- GCF-036323735.1, rat
- GCF-028858775.2, Chimpanzee
- GCF-018350175.1, cat
- GCF-011100685.1, dog
- GCF-000696695.1, burgud
- GCF-000003025.6, pig
- GCF-016772045.2, sheep
- GCF-016699485.2, chicken
- GCF-025200985.1, fly
- GCF-000002035.6, Zebrafish
- GCF-029289425.2, Pygmy Chimpanzee
- GCF-029281585.2, Western gorilla
- GCF-028885655.2, Pongo abelii
- GCF-028885625.2, Bornean orangutan
- GCF-003339765.1, Rhesus monkey
- GCF-037993035.1, Macaca fascicularis
- GCF-003668045.3, Chinese hamster

---
**Algorithm 2** Differential Classification Wrapper
---
1: **procedure** INITIALIZEWRAPPER($\mathcal{M}_{\text{base}}$, $C$, decoder, usePooler)
2:    $d \leftarrow \mathcal{M}_{\text{base}}.\text{hidden\_size}$                                                                           ▷ fallback to $d_{\text{model}}$ if absent
3:    **if** usePooler **then**
4:       Pooler $\leftarrow \text{Linear}(d, d) \circ \tanh$
5:    **else**
6:       Pooler $\leftarrow$ identity
7:    **end if**
8:    Head $\leftarrow \text{Linear}(d, C)$
9:    **return** $\mathcal{W} = \{\mathcal{M}_{\text{base}}, \text{Pooler}, \text{Head}, \text{decoder}\}$
10: **end procedure**
11:
12: **procedure** FORWARD($\mathcal{W}$, $\mathbf{x}^{\text{ref}}$, $\mathbf{m}^{\text{ref}}$, $\mathbf{x}^{\text{alt}}$, $\mathbf{m}^{\text{alt}}$, $y$)
13:    $\mathbf{H}^{\text{ref}} \leftarrow \mathcal{M}_{\text{base}}(\mathbf{x}^{\text{ref}}, \mathbf{m}^{\text{ref}}).\text{lastHidden}$
14:    $\mathbf{H}^{\text{alt}} \leftarrow \mathcal{M}_{\text{base}}(\mathbf{x}^{\text{alt}}, \mathbf{m}^{\text{alt}}).\text{lastHidden}$
15:    **if** $\mathcal{W}.\text{decoder}$ **then**
16:       $\mathbf{h}^{\text{ref}} \leftarrow \text{LASTTOKEN}(\mathbf{H}^{\text{ref}}, \mathbf{m}^{\text{ref}})$
17:       $\mathbf{h}^{\text{alt}} \leftarrow \text{LASTTOKEN}(\mathbf{H}^{\text{alt}}, \mathbf{m}^{\text{alt}})$
18:    **else**
19:       $\mathbf{h}^{\text{ref}} \leftarrow \mathbf{H}^{\text{ref}}[:, 0]$                                                                 ▷ [CLS] token
20:       $\mathbf{h}^{\text{alt}} \leftarrow \mathbf{H}^{\text{alt}}[:, 0]$
21:    **end if**
22:    $\mathbf{h}^{\text{ref}} \leftarrow \text{Pooler}(\mathbf{h}^{\text{ref}})$
23:    $\mathbf{h}^{\text{alt}} \leftarrow \text{Pooler}(\mathbf{h}^{\text{alt}})$
24:    $\Delta\mathbf{h} \leftarrow \mathbf{h}^{\text{alt}} - \mathbf{h}^{\text{ref}}$
25:    $\hat{y} \leftarrow \text{Head}(\Delta\mathbf{h})$
26:    **if** $y \neq \varnothing$ **then**
27:       $\mathcal{L} \leftarrow \text{CrossEntropy}(\hat{y}, y)$
28:       **return** $\{\mathcal{L}, \hat{y}\}$
29:    **else**
30:       **return** $\hat{y}$
31:    **end if**
32: **end procedure**
---

- GCF-041296265.1, horse

The construction process follows these key steps:

**Sequence Extraction and Functional Annotation**

- The raw dataset includes a diverse set of RNA sequences with different functional annotations.
- The original dataset consists of the following RNA types and their counts:
    - mRNA: 220397
    - tRNA: 20674
    - snRNA: 18996
    - snoRNA: 15577
    - lncRNA: 14291
    - miRNA: 12238
    - primary-transcript: 8644
    - rRNA: 6225
    - transcript: 3860
    - ncRNA: 477
    - guide-RNA: 143
    - antisense-RNA: 28

- RNase-P-RNA: 9
- V-gene-segment: 7
- scRNA: 7
- Y-RNA: 6
- telomerase-RNA: 5
- vault-RNA: 4
- SRP-RNA: 4
- RNase-MRP-RNA: 3
- C-gene-segment: 2
- D-gene-segment: 1

- To refine the dataset, only **seven functional RNA types** are retained:
  - mRNA
  - tRNA
  - snRNA
  - snoRNA
  - lncRNA
  - miRNA
  - rRNA

**Annotation Enhancement Using a Large Language Model (LLM)**

- A **LLM Annotator** is employed to extend and refine functional annotations.
- The model is prompted with: *Describe the functions of the given gene/RNA precisely and concisely."*
- For example, an initial annotation like Lnc RNA 2789" is expanded into:

    "It is Lnc RNA 2789 that doesn't encode proteins but regulates various biological processes."

**Final Dataset Composition**

- The final dataset consists of **300,000 DNA-function pairs**, where each DNA sequence is paired with an enhanced functional description.
- These functionally annotated sequences serve as high-quality input for machine learning models.

| Usage | Prompt |
|---|---|
| Used by **LLM (Annotator)** for extending the short annotation to detailed function annotation | **(System Prompt)** You are a helpful assistant that answers functions of given gene/rna precisely and concisely. |
| Used by **LLM (Evaluator)** for deciding the type of DNA given its function description | **(System Prompt)** You are a helpful assistant that determines the type of RNA based on the given function description. When deciding the functoin. Your answer should to only be one of the ['mRNA', 'tRNA', 'snRNA', 'snoRNA', 'lncRNA', 'miRNA', 'rRNA'] |
| Given to **GPT4o@zeroshot** for performing zeroshot DNA function prediction | **(System Prompt)** You are a helpful assistant that determines the type of RNA based on the given DNA sequence. Your answer should only be one of the mRNA, tRNA, snRNA, snoRNA, lncRNA, miRNA, rRNA |

Table 9: Prompt Used during dataset construction, evaluation and zeroshot prediction

| Ground Truth | GPT4o | OmniDNA@ft |
|---|---|---|
| **(Input)** TGGCAGAGATATG...AAGAAAAAAGAAAATGAACTTGGTGCAGGCA (DNA of length 1000) | | |
| mRNA:CD209, also known as DC-SIGN (Dendritic Cell Specific Intercellular adhesion molecule-3-grabbing Noninte-grin), is a type I trans-membrane protein primarily expressed on dendritic cells and some other immune cells. It functions as a receptor that binds and internalizes pathogens, including viruses and bacteria, thereby facil-itating their presentation to `T cells` and playing a critical role in the `immune response`. CD209 is involved in the recog-nition of mannose-rich lig-ands, mediating cellular adhe-sion, and influencing various aspects of `immune activation` and tolerance. Additionally, it can `modulate inflammatory` responses and has been im-plicated in HIV-1 and other pathogen infections. | The provided DNA sequence likely represents a genomic re-gion involved in regulatory or coding functions. Based on its composition, it may con-tain promoter elements, cod-ing regions, untranslated re-gions (UTRs), or repetitive se-quences. | C-C motif chemokine lig-and 22 ( CCL22 ) is a chemokine involved in immune responses . It primarily at-tracts and `activates T cells` , particularly regulatory T cells ( Tregs ), and plays a role in modulating inflammation and `immune responses`. CCL22 is produced by various cell types , including macrophages and dendritic cells , and is im-plicated in various diseases , in-cluding autoimmune disorders and cancer. |

Table 10: Comparison of Ground Truth, GPT4o, and OmniDNA@ft in RNA type determination (Example 1)

# G   Attention Sink Visualization

To investigate the emergence of attention sinks in our model, we visualize the self-attention scores across several transformer layers during inference. Specifically, we load the `Omni-DNA-116M` model and generate a random DNA sequence of length 100. This sequence is tokenized and passed through the model without applying a causal attention mask, as many downstream genomic tasks require bidirectional context.

We attach an attention probe to each transformer block, which records the softmax-normalized attention weights during the forward pass. In Figure 7, we show the attention matrices for four layers (`L0, L1, L-2, L-1`), each across four attention heads.

The visualization reveals distinct patterns: in `Layer 1`, the attention is diffuse without a clear sink; in `Layer 2`, we observe a strong attention sink at the `START` token; and in the last two layers, both the `START` and `END` tokens receive significant attention, suggesting the model learns to aggregate global information around these special positions in deeper layers.

# H   Additional Ablation Study On SeqPack Algorithm

**Additional Ablation on the SeqPack Parameter with Different Compression Ratios**    To ablate the parameter of **SeqPack** compression ratios ($c = \frac{L_{original}}{L_{pruned}}$) and assess SeqPack's effectiveness against established methods, we adapted *Dynamic Context Pruning (DCP)* [1] to work with **Omni-DNA 116M** on the *Enhancer–Target Prediction* task. The results demonstrate SeqPack's substantial advantages.

| Ground Truth | GPT4o | OmniDNA@ft |
|---|---|---|
| (Input) GAGGTAGTAGATTGTATAG | | |
| miR -3596d is a microRNA found in rats (Rattus norvegicus). It is involved in the regulation of gene expression at the post-transcriptional level, primarily by binding to complementary sequences in messenger RNAs (mRNAs), leading to their degradation or inhibition of translation . While specific target genes may vary, miRNAs like rno-miR-3596d generally play roles in various biological processes such as development, cell differentiation, proliferation, and apoptosis. Specific roles or targets for rno-miR-3596d would require experimental validation or further studies to elucidate its functions in rat biology. | The DNA sequence GAG-GTAGTAGATTGTATAG could serve various functions depending on its genomic context. Generally, it could be: 1. A regulatory element – If found in promoter or enhancer regions, it may influence gene expression. 2. A binding site – It could be a recognition site for transcription factors, such as ETS-family proteins (which recognize GGAA/T motifs). 3. A coding or non-coding region – If within a gene, it might encode a peptide or function as a non-coding RNA. 4. A structural or repeat element – It could contribute to chromatin organization or genome stability.For precise functional annotation, its location within a genome and experimental validation would be needed. | gga - miR - 65 77 - 5p is a microRNA ( miRNA ) found in chickens ( Gallus gallus ). MicroRNAs are small , non - coding RNA molecules that play a crucial role in regulating gene expression by binding to complementary sequences on target messenger RNAs ( mRNAs ), leading to their degradation or inhibition of translation . Specifically , gga - miR - 65 77 - 5p may be involved in various biological processes , including development , differentiation , and response to stress , although specific target genes and functions may require further study |

Table 11: Comparison of Ground Truth, GPT4o, and OmniDNA@ft in RNA type determination (Example 2)

At extreme compression ratios (220×, 150×), DCP performance degrades severely, barely outperforming random baselines, while SeqPack maintains strong performance (AUROC > 0.89). Even at moderate compression (50×), SeqPack outperforms DCP (0.907 vs. 0.880 AUROC).

**The Computational Overhead of SeqPack**  Compressing inputs (25,600–819,200 tokens) to a fixed 4,096 with SeqPack, we report (i) SeqPack operator time and (ii) Omni-DNA-1B forward time in seconds on the compressed sequence. As input length grows 32×, SeqPack time rises only 3.56× (inference: 0.00898 to 0.03197 s) and 5.95× (training: 0.015 to 0.091 s). Forward time remains effectively flat. SeqPack's share of end-to-end time remains low and increases gently with the compression ratio (6.25× to 200×): 1.2% to 4.0% (inference) and 4.3% to 21.6% (training).

| Ground Truth | GPT4o | OmniDNA@ft |
|---|---|---|
| **(Input)** TGCTCACTTCAGCAGCACATGTGCTAAAAT TGGAATGACACAAAGAAGATTAGCATGGCCGC TGAGCAAGGATGACACGCACATTTGTGAATTGTTCCATAAAAAA | | |
| nRNA:U6 spliceosomal RNA is a small nuclear RNA (snRNA) that plays an essential role in the splicing of pre-mRNA in eukaryotic cells. It is a core component of the spliceosome, the complex responsible for the removal of introns from pre-mRNA transcripts. U6 engages in base-pairing interactions with other spliceosomal RNAs and pre-mRNA, facilitating the assembly and function of the spliceosome. Its primary roles include catalyzing the cleavage of introns and the ligation of exons during the splicing process. U6 is also involved in the activation of the catalytic core of the spliceosome, making it crucial for proper gene expression. | The given DNA sequence likely encodes a functional genetic element, such as a protein-coding region or a regulatory sequence. To determine its precise function, I will need to analyze it for potential coding regions, motifs, or regulatory elements. | U6 spliceosomal RNA is a small nuclear RNA ( snRNA ) that plays a crucial role in the splicing of pre - mRNA in eukaryotic cells . It is a component of the spliceosome , the complex responsible for removing introns from pre - mRNA and joining exons together . U6 RNA is involved in the catalytic activity of the spliceosome and is essential for the recognition of splice sites and the formation of the active site for splicing . It also interacts with other snRNAs. |

Table 12: Comparison of Ground Truth, GPT4o, and OmniDNA@ft in RNA type determination (Example 3)

Table 13: Performance comparison between SeqPack and DCP under various compression ratios on the Enhancer–Target Prediction task.

| Model | AUROC ($\pm$ SE) | Compression Ratio |
|---|---|---|
| ABC (expert) | 0.926 | — |
| CNN | $0.803 \pm 0.022$ | — |
| SeqPack | $0.894 \pm 0.015$ | 220 |
| SeqPack | $0.899 \pm 0.009$ | 150 |
| SeqPack | $0.907 \pm 0.011$ | 50 |
| DCP | $0.502 \pm 0.089$ | 220 |
| DCP | $0.632 \pm 0.010$ | 150 |
| DCP | $0.880 \pm 0.012$ | 50 |

Table 14: Inference-time and Training-time analysis of SeqPack compression across varying input lengths. For inference, SeqPack operator time grows sublinearly with sequence length, while forward time remains stable. At training time, SeqPack accounts for an increasing but still moderate fraction of total time as sequence length increases.

| Length | SeqPack (s) | Forward (s) | SeqPack % |
|---|---|---|---|
| 25,600 | 0.00898 ($\pm$0.000) | 0.77257 ($\pm$0.003) | 1.1 |
| 51,200 | 0.01032 ($\pm$0.001) | 0.76120 ($\pm$0.001) | 1.3 |
| 102,400 | 0.01225 ($\pm$0.001) | 0.77986 ($\pm$0.003) | 1.5 |
| 204,800 | 0.01508 ($\pm$0.001) | 0.76666 ($\pm$0.001) | 1.9 |
| 409,600 | 0.02050 ($\pm$0.001) | 0.76665 ($\pm$0.001) | 2.6 |
| 819,200 | 0.03197 ($\pm$0.001) | 0.76514 ($\pm$0.003) | 4.0 |

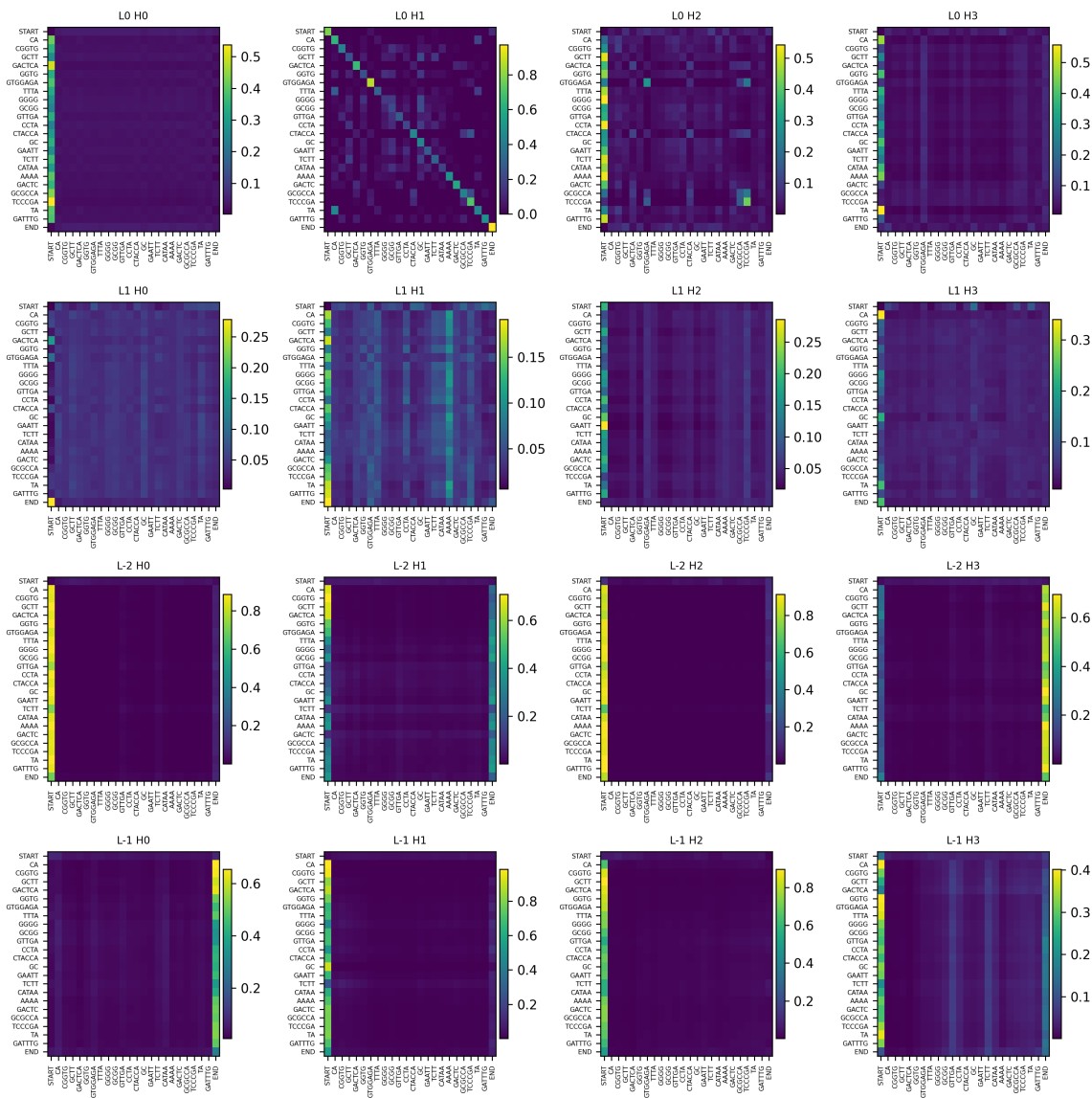

Figure 7: **Attention patterns across layers and heads in Omni-DNA.** Each row corresponds to a different transformer layer (L0, L1, L-2, L-1), and each column to a different attention head (H0–H3). For layer 1, attention is relatively diffused with no clear sink. At layer 2, we observe a strong attention sink at position 0 (START). In the last two layers, both the START and END tokens exhibit sink-like behavior, suggesting global aggregation mechanisms emerging in deeper layers.

| Length | SeqPack (s) | Forward (s) | SeqPack % |
|---|---|---|---|
| 25,600 | 0.01523 ($\pm$0.000) | 0.33756 ($\pm$0.002) | 4.3 |
| 51,200 | 0.01793 ($\pm$0.000) | 0.32607 ($\pm$0.002) | 5.2 |
| 102,400 | 0.02339 ($\pm$0.001) | 0.34088 ($\pm$0.001) | 6.4 |
| 204,800 | 0.03389 ($\pm$0.001) | 0.32727 ($\pm$0.002) | 9.4 |
| 409,600 | 0.05290 ($\pm$0.001) | 0.32809 ($\pm$0.002) | 13.9 |
| 819,200 | 0.09067 ($\pm$0.001) | 0.32850 ($\pm$0.002) | 21.6 |

