# OpenReview forum: "Omni-DNA: A Genomic Model Supporting Sequence Understanding, Long-context, and Textual Annotation"
_NeurIPS.cc/2025/Conference — NeurIPS 2025 poster_

### Official Review · Reviewer_MarC · 2025-06-27

**Clarity:** 4
**Significance:** 3
**Originality:** 3
**Rating:** 5
**Confidence:** 3

**Summary:**

This paper presents a new family of genomic foundation models, OmniDNA. The authors conduct extensive comparisons against widely used baselines on several downstream tasks. The authors also present a novel algorithm, Seqpack, for extending context length of existing pre-trained models.

**Questions:**

1. The authors discuss several approaches for extending pre-training vocabularies (Lines 142-149). Which approach worked best / is used in the experiments?
2. The motivation for SeqPack is that $O(N \log K)$ is too expensive. Ultimately SeqPack costs $\Theta(N)$, is the savings here indeed that significant? I would have thought the $\log K$ factor would not be that untenable?
3. In Section 3.3.1, the gumbel softmax trick is used to maintain differentiability, but it is then composed with top-k selection. Doesn’t applying top-k destroy differentiability?
4. Can the authors confirm that all the models from Section 4.2 (Table 4) have 450kbp input sequences. Additionally, would it be possible to provide error bars for Table 4?
5. What is the positive class ratio of the final datasets used in the experiments from Section 4.3.

**Ethical Concerns:**

["NO or VERY MINOR ethics concerns only"]

**Final Justification:**

Authors addressed my concerns and provided extensive additional experimental results

**Limitations:**

See weaknesses.

**Quality:**

4

**Strengths And Weaknesses:**

### **Strengths**

The experimental work in this paper is extensive and the consistently superior results of the OmniDNA models are promising. The novel Seqpack algorithm is a nice contribution and the ablations on mult-task training and the positional embeddings are useful for practitioners in the field. The breadth of baselines is well curated and the experimental details provided to reproduce the experiments are well organized and extensive. The natural language interaction tasks are novel and interesting.

### **Weaknesses**

1. I disagree with the framing of several of the limitations and contributions.
    - Specifically, claiming that there has not been exploration of the design space (Limitation 1) seems inaccurate to me. Several models have explored the effect of tokenization (e.g., DNABERT 1 vs 2, and the base pair tokenization of Hyena and Caduceus) and have employed different positional embedding strategies (e.g., NTv1 vs NTv2). Additionally, Caduceus, in particular, created custom extensions of Mamba-based modules specifically for genomic tasks. The authors claim that rather than the “plug-and-play” of existing NLP modules from existing GFMs, they optimize their backbone for genomics. However, I think one could argue that they performed a search over the same modules that other GFMs have employed. I believe that a reframing would be more accurate, improve readability, and would by no means detract from the extensive experiments and useful results of Finding 1 in the paper.
    - In a similar vein, I disagreed somewhat with the motivation of limitation 2, in the sense that arguing for the need of multi-task training from the lens of expensive storing of weights is not reflective of the reality of how these models will be used or the true bottlenecks that practitioners face with the current scale of GFMs. Here too, more accurately situating the experiments and results of Finding 3 in the rich literature of multi-task training (even in the context of biological models, e.g. Enformer) would be better, in my opinion. This by no means takes away from the fact that exploring multi-task downstream fine-tuning of GFMs is indeed a contribution of this work.
    - Lastly for limitation 4, the authors state that previous GFMs cannot adapt to varying context lengths, but this claim needs to be supported either with references from the literature or by pointing to specific experimental results from this work that support the claim.
2. The takeaways from Finding 2 are unclear to me. Several works have already indicated that the NT tasks are in fact not ideal for understanding GFMs (e.g., Kirill, et al [1]). Additionally, it is possible that these tasks are too “easy” and can be saturated. Lastly, unless I am misreading it, I think Figure 3 does in fact show an important correlation between pre-training loss and downstream performance.
3. I think the authors could include a more robust discussion about what is making OmniDNA work. For example:
    - Is data curation an important factor?
    - Is the fact that the authors are using a transformer-based AR model vs BERT-style bi-directional one important (e.g., NTv2 and DNABERT are transformers but not AR, Hyena is AR but not a transformer; so is the claim of OmniDNA that the correct recipe is AR + transformer)?
    - Is the combination of positional embedding + layer norm a key driver? Specifically, these design decisions are already explored / used in previous models (e.g. NTv2 uses RoPE).
4. An ablation exploring the efficacy of SeqPack vs existing context extension mechanisms or even the naive approach of simply providing pre-trained models with longer sequences would be useful and convincing. Additionally, exploring the hyperparameters of SeqPack would be informative
5. Additionally, the computational overhead of SeqPack is not well described or quantified. Adding this would be useful.
6. Some minor typos:
    - Line 174: “Understand the…” should be “To understand the…”?
    - Line 235: “strictly output” should be “strictly outperform”?


---

[1] Vishniakov, Kirill, et al. "Genomic foundationless models: Pretraining does not promise performance." bioRxiv (2024): 2024-12.

---

> ### Author Rebuttal · Authors · 2025-07-31
>
> We sincerely appreciate your detailed feedback and constructive remarks on our paper. Below, we provide a detailed response to your questions and comments, along with additional ablation results from your suggestion.
>
> ### Q1. The authors discuss several approaches for extending pre-training vocabularies (Lines 142-149). Which is used in the experiments?
>
> There are two tasks which introduce new vocabulary to the Omni-DNA: Next-Token Prediction approach for 10 tasks (section 4.1 Multitasking) and Sequence-to-Function Generation. The key-token replication is only used for multi-tasking; while NEFTune (i.e., adding isotropic Gaussian noise) is used for both tasks.
>
> ### Q2. ...the $logK$ factor would not be that untenable?
>
> While the theoretical difference between $O(N\log K)$ and $\Theta(N)$ might appear marginal, the practical efficiency gains of our method are impactful due to the following reasons:
>
> - 1. As SeqPack needs to be integrated into both finetuning and inference, when N is very large, logK*N could lead to differences as well.
> - 2. Memory footprint is another concern. When we perform global sampling, we need to maintain relatively large tensors in memory, while the splitting approach allows the sequential/parallel processing of each segment.
> - 3. Implicit prior, by encouraging the compressor to sample from each segment, we are enforcing an implicit uniform prior, which could help to avoid the potential issue of collapsing the learned mass on a single region.
>
>
> ### Q3. Doesn’t applying top-k destroy differentiability?
>
> If we directly use the top-K selection method, we get hard binary choices (either a token is selected or not) and this creates a problem: since the selection is discrete, there's no gradient information flowing back through the selection process. We use Gumbel–Softmax to build a soft, differentiable selector $y_{\text{soft}}$ from the scores, then apply a straight-through estimator: the forward pass uses hard one-hot indices for behavior, while the backward pass uses the soft probabilities for gradients. Thus, the model learns from a smooth relaxation even though the forward selection is discrete. At inference, we switch to ordinary discrete selection (sampling or top-k).
>
>
> ```
> // Inputs
> // X: (B, L, D) token embeddings
> // weights: (L,) non-negative scores (normalized internally)
> // k: number of tokens to keep
> OUTPUT: selected_token_embeddings(B,k,D)
>
> IF training_mode:
>     // Gumbel-Softmax for differentiable selection
>     logits = expand(log(weights), to=(k,L))
>     gumbel = sample_gumbel_noise(shape=(k,L))
>     y_soft = softmax(logits + gumbel )
>
>     // Straight-through: hard forward, soft backward
>     y_hard = one_hot(argmax(y_soft))
>     y_st = detach(y_hard - y_soft) + y_soft
>
>     // Apply selection and return
>     RETURN einsum('kl,lbd->kbd', y_st, transpose(sequence))
> ELSE:
>     // Standard multinomial sampling for inference
>     indices = multinomial(weights, k, no_replacement=True)
>     RETURN sequence[:, sort(indices), :]
>
> ```
>
>
> ### Q4. Can the authors confirm that all the models from Section 4.2 (Table 4) have 450kbp input sequences. Additionally, would it be possible to provide error bars for Table 4?
> Yes, we confirm that all the models are evaluated with the input sequence of length 450,000 bp. We use the original benchmark from [1].
> Following your suggestion, we ran the code from the original benchmark to obtain the new table with error bars as shown below, and we will integrate this into the paper.
>
>
> | Model        | AUROC (± SE) |
> |--------------|-------------|
> | ABC (expert) | 0.926  |
> | CNN          | 0.803 ± 0.022 |
> | HyenaDNA     | 0.826 ± 0.014 |
> | CADUCEUS-PH  | 0.821 ± 0.023 |
> | CADUCEUS-PS  | 0.814 ± 0.020 |
> | **Omni-DNA** | **0.894 ± 0.015** |
>
>
> ### Q5.What is the positive class ratio of the final datasets used in the experiments from Section 4.3.
>
> The positive: negative ratio is 1:1.
>
>
> ### Q6. Additional ablation on the SeqPack Parameter with different Compression Ratio
>
>
> To abalate the parameter of SEQPACK compression ratios (c = $\frac{L_{original}}{L_{pruned}}$) and assess SEQPACK's effectiveness against established methods, we adapted Dynamic Context Pruning (DCP)[2] to work with Omni-DNA 116M on the Enhancer-Target Prediction task. The results demonstrate SEQPACK's substantial advantages.
>
>
> | Model        | AUROC (± SE) | Comp. Ratio |
> |--------------|--------------|-------------|
> | ABC (expert) | 0.926  | — |
> | CNN          | 0.803 ± 0.022 | — |
> | **Seq-Pack** | 0.894 ± 0.015 | 220 |
> | **Seq-Pack** | 0.899 ± 0.009 | 150 |
> | **Seq-Pack** | **0.907 ± 0.011** | 50 |
> | **DCP** | 0.502 ± 0.089 | 220 |
> | **DCP** | 0.632 ± 0.010 | 150 |
> | **DCP** | 0.880 ± 0.012 | 50 |
>
> At extreme compression ratios (220×, 150×), DCP performance degrades severely, barely outperforming random baselines, while SEQPACK maintains strong performance (AUROC > 0.89). Even at moderate compression (50×), SEQPACK outperforms DCP (0.907 vs 0.880 AUROC).
>
> ### Q7. The computational overhead of SeqPack
>
>
> Compressing inputs (25,600–819,200 tokens) to a fixed 4,096 with SeqPack, we report (i) SeqPack operator time and (ii) Omni-DNA-1B forward time in seconds on the compressed sequence. As input length grows 32×, SeqPack time rises only 3.56× (inference: 0.00898 to 0.03197 s) and 5.95× (training: 0.015 to 0.091 s). Forward time is effectively flat. SeqPack’s share of end-to-end time remains low and increases gently with the compression ratio 6.25× to 200×: 1.2% to 4.0% (inference) and 4.3% to 21.6% (training).
>
> **Inference**
> | length | SeqPack  |  Forward  | SeqPack_percent% |
> |---:|---:|---:|---:|
> | 25,600 | 0.00898 (±0.000) | 0.77257 (±0.003) | 1.1 |
> | 51,200 | 0.01032 (±0.001) | 0.76120 (±0.001) | 1.3 |
> | 102,400 | 0.01225 (±0.001) | 0.77986 (±0.003) | 1.5 |
> | 204,800 | 0.01508 (±0.001) | 0.76666 (±0.001) | 1.9 |
> | 409,600 | 0.02050 (±0.001) | 0.76665 (±0.001) | 2.6 |
> | 819,200 | 0.03197 (±0.001) | 0.76514 (±0.003) | 4.0 |
>
> **Training**
> | length | SeqPack  |  Forward  | SeqPack_percent% |
> |---:|---:|---:|---:|
> | 25,600 | 0.01523 (±0.000) | 0.33756 (±0.002) | 4.3 |
> | 51,200 | 0.01793 (±0.000) | 0.32607 (±0.002) | 5.2 |
> | 102,400 | 0.02339 (±0.001) | 0.34088 (±0.001) | 6.4 |
> | 204,800 | 0.03389 (±0.001) | 0.32727 (±0.002) | 9.4 |
> | 409,600 | 0.05290 (±0.001) | 0.32809 (±0.002) | 13.9 |
> | 819,200 | 0.09067 (±0.001) | 0.32850 (±0.002) | 21.6 |
>
> ### Q8. Additional Ablations on why Omni-DNA works
> We generally believe the transformer's learning capability is the reason why the model works. To see the effect of the training paradigm (BERT vs. NextTokenPrediction), the additional ablations are conducted, with detailed results presented in our response to Q1 of reviewer 1(rvnr).  We could see that the BERT pretraining, in terms of the data source, we found that the number of training samples is a key. But NTP training provides the model with the natural ability for sequence generation, which is generally preferred and has substantial downstream use cases. This is the reason why we use the NTP as the default training scheme. But due to the limit of computational resources, we may not be able to explore all the architectural choices.
>
> ### Q9.  Finding 2
> Finding 2 is to emphasize that the lower loss **does not guarantee** better performance in downstream tasks. We can see several bumps in Figure 3, which means such relations are not guaranteed. We should be cautious about using the loss as the only criterion for selecting the best model.
>
> ### Q10. Regarding a few suggestions of the introduction section.
>
> We appreciate your kind suggestion. The below is the reframed paragraph to replace line 31-40 of the original manuscript.
>
> While GFMs have demonstrated utility on specific genomic tasks, they still exhibit important limitations: **(i) Limited optimization of architectural components for genomic tasks:** Prior work has explored design choices including tokenization (DNABERT vs. DNABERT-2), positional encodings (NT vs. NTv2), and genomic-specific modules (GPN-MSA, Caduceus), but a more comprehensive, task-specific search of architectures and pretraining strategies tailored to genomic inductive biases is still lacking. **(ii) Limited exploration of multi-task synergies:** Many approaches fine-tune on a single downstream task, missing the opportunity to exploit shared biological signal and achieve cross-task performance gains, despite evidence of such benefits in genomic assay prediction (e.g., AlphaGenome, Enformer). **(iii) Insufficient multimodal capability:** Most existing GFMs restrict vocabularies to nucleotide base pairs, predefined labels, or learned subwords with the BPE tokenizer, limiting tasks that require mapping DNA sequences to functional annotations in natural language (DNA2Func). While ChatNT can process (DNA, text) inputs, it uses separate tokenizers and a two-stage stack consisting of NTv2 as the DNA encoder and Vicuna-7B as the decoder, resulting in less unified handling of multimodal information. **(iv) Context-length adaptivity challenges:** Transformer-based models (e.g., DNABERT-2, NTv2) remain constrained by the quadratic cost of self-attention. This limits the application of transformer-based models to long-context tasks such as enhancer–target prediction.
>
> [1] Cheng, Wenduo, et al. "Dnalongbench: a benchmark suite for long-range dna prediction tasks." bioRxiv 2025.
>
> [2] Anagnostidis, Sotiris, et al. "Dynamic context pruning for efficient and interpretable autoregressive transformers." NeurIPS 2023.

---

> > ### Comment · Reviewer_MarC · 2025-08-01
> >
> > Thank you for the detailed response and the additional experimental results.
> >
> > The authors have sufficiently addressed my concerns and I will raise my score to 5 accordingly.
> >
> > My one remaining contention is with the experiments for Finding 2: I think these results are more a comment on the nature of the NT and GenomicsBenchmarks task as opposed to the usefulness of pre-training loss as a signal for downstream performance, as the other works have also found these benchmarks to be lacking and there is very strong evidence from prior works (especially if we also consider other fields, such as NLP) that minimizing perplexity is a key driver of downstream performance.

---

> > > ### Author Response · Authors · 2025-08-01
> > >
> > > Thank you for the prompt follow-up and positive feedback. Regarding Finding 2, we agree with your assessment. In the revision, we will explicitly limit the claim to our setting—sequence-classification tasks drawn from NT and GenomicsBenchmarks—and avoid generalizing beyond these datasets.

---

> > > > ### Comment · Reviewer_MarC · 2025-08-04
> > > >
> > > > Thank you for your open-mindedness to feedback here.

---

### Official Review · Reviewer_UqSE · 2025-07-02

**Clarity:** 3
**Significance:** 4
**Originality:** 3
**Rating:** 5
**Confidence:** 4

**Summary:**

This paper presents Omni-DNA, a new autoregressive DNA language model that includes a novel method of long context compression. The model is trained on genome assemblies from NCBI. It is fine-tuned by adding additional tokens or multi-head fine-tuning. It is benchmarked on Gene regulatory element classification, Pathogenic variant effect prediction, Enhancer–target gene interaction mapping, and Sequence-conditioned functional interpretation generation.

**Questions:**

1. For enhancer-target prediction, did you mask the intervening bases between the enhancer and the promoter as in (https://openreview.net/forum?id=opv67PpqLS)? If so, how was this masking accomplished?

2. I am very confused by the genetic disease variant-type classification. Conventionally, models are evaluated by their ability to classify disease-causing variants from negative control variants, which have no phenotypic effect. The assumption is that the models should be able to identify variants that change protein sequence or alter conserved regulatory sequences, vs. variants that have no impact. Here, the study seems to have taken the surprising route of classifying variants that cause one disease from variants that cause a different disease. All of these variants are pathogenic, so what is the model classifying exactly? My guess is that it is learning the genes near the variants, and has learned that some genes are associated with a particular disease - in which case it is not really evaluating whether the model has learned properties of variants. Please clarify the purpose of this task or draw the negative controls from non-pathogenic variants.

**Ethical Concerns:**

["NO or VERY MINOR ethics concerns only"]

**Final Justification:**

I am satisfied with the explanation of masking strategy.

I remain unconvinced by the evaluation of sequence models based on their ability to link pathogenic variants to a disease - it still seems to me that this is likely to reflect memorization that certain regions of the genome are disease-associated, rather than the model understanding the properties of the variants themselves (e.g. predicting their effect on gene expression). However, seeing that the authors have added a non-pathogenic variant baseline where Omni-DNA also shows the highest performance addresses this comment.

As such, I maintain the previous rating and the recommendation to accept.

**Limitations:**

yes

**Quality:**

3

**Strengths And Weaknesses:**

The model achieves SOTA performance on classification tasks which is impressive. The task of generating textual descriptions of DNA sequences also appears very useful and original, adding a novel capability to the current foundation models in the field. However, the remaining evaluations are confusing and could use some clarification or editing (see below).

---

> ### Author Rebuttal · Authors · 2025-07-31
>
> Thank you for your positive feedback of and constructive suggestion for our manuscript. Below, we address each of your questions and comments in detail.
>
> ### Q1. Did you mask the intervening bases between the enhancer and how was this masking accomplished?
> We adopt the curated benchmark from Cheng et al. [1] and use the same masking strategy to mask the intervening enhancers. In the original implementation, the positions of the intervening enhancers in K562 cell lines are recorded in a text file as a blacklist. This file includes the chromosome, start, end, and strand of the enhancers that need to be masked. During data loading with selene_sdk (a python library), the masked positions are replaced with [0.25, 0.25, 0.25, 0.25]. In our case, we replace these positions with 'N' for Omni-DNA to consume.
>
>
> ### Q2. Genetic disease variant-type classification
>
> **Objective:** Conventional training on benign vs. pathogenic labels is valuable but inherently coarse: it often fails to capture nuanced mutation mechanisms—such as how mutation direction varies across diseases and how these differences map to distinct disease phenotypes within the same gene. Our objective in this task is to move beyond generic assessments of deleteriousness and enable the model to learn disease-related patterns that reflect fine-grained functional changes. Disease-type labels serve as a form of weak supervision for learning mutation directionality—something that a simple benign-vs-pathogenic objective cannot reliably disentangle.
>
> In real-world workflows (e.g., cardiac clinics), clinicians need to prioritize variants specific to the target disease in the form of patient reports, rather than flagging all pathogenic variants across conditions. Current practice relies heavily on rule-based gene–disease associations from resources such as OMIM[2] and the Clinical Genomic Database (CGD)[3] as disease-focused filters. We demonstrate that by fine-tuning general sequence models with disease-type labels, we can rapidly recover these associations and highlight the most relevant variants for the disease of interest—thereby aligning model outputs with established clinical triage pipelines.
>
>
> **Draw the negative controls from non-pathogenic variants:** Nevertheless, we are happy to include the additional results in the appendix, based on sampling negatives from non-pathogenic variants, as shown below. Specifically, we sample the negatives from benign variants with an AF_TGP (allele frequency from the 1000 Genomes Project) less than 0.01, using the same version of ClinVar. The classification F1 is generally higher than that of the disease-type classification task (as shown in Table 5 of the manuscript).
>
>
> | Task | HyenaDNA | DNABERT2 | NTv2 | Omni-DNA |
> |---|---:|---:|---:|---:|
> | **LC** | 0.85 ± 0.02 | 0.73 ± 0.03 | 0.77 ± 0.04 | 0.92 ± 0.01 |
> | **ICCs** | 0.80 ± 0.08 | 0.62 ± 0.02 | 0.76 ± 0.00 | 0.91 ± 0.01 |
>
> **What does the model learn:**
> In order to examine whether the model learns disease–variant associations by memorizing gene identity or other non-coding regions, we performed a simple ablation study for lung cancer. In this setup, genes were masked with 'N' (based on the human gene annotation file), and the model was still able to achieve reasonable performance. Note that implicitly including gene position is not necessarily undesirable, as this aligns with how hospitals typically prioritize variants using rule-based systems.
>
> | Model      | HyenaDNA     | DNABERT2     | NTv2         | Omni-DNA     |
> |------------|--------------|--------------|--------------|--------------|
> | LC         | 0.83±0.02    | 0.71±0.03    | 0.75±0.04    | 0.93±0.01    |
> | LC (MaskGene)   | 0.79±0.01    | 0.60±0.04    | 0.68±0.02    | 0.87±0.00|
>
>
>
>
> [1] Cheng, Wenduo, et al. "Dnalongbench: a benchmark suite for long-range dna prediction tasks." bioRxiv 2025
>
> [2] https://www.omim.org/  OMIM (Online Mendelian Inheritance in Man)
>
> [3] https://research.nhgri.nih.gov/CGD/  CGD (Clinical Genomic Database)

---

> ### Author Response · Authors · 2025-08-04
>
> Thank you again for your constructive feedback on our submission. We're writing to follow up with the rebuttal. If there is any follow-up questions, we are happy to answer.

---

> > ### Comment · Reviewer_UqSE · 2025-08-05
> >
> > Thank you. My concerns have been addressed. I am still uncertain about what exactly the model is learning in the disease classification task, but I am satisfied with the inclusion of the evaluation with non-pathogenic variants. I have maintained my original rating.

---

### Official Review · Reviewer_nFBa · 2025-07-02

**Clarity:** 1
**Significance:** 2
**Originality:** 3
**Rating:** 4
**Confidence:** 3

**Summary:**

This manuscript introduces the Omni-DNA. a family of models spanning 20M to 1.1B parameters that supports sequence understanding, long-context genomic reasoning, and natural-language annotation.

Omni-DNA is a decoder only structure for modelling genetic information. The main contribution of this manuscript lie in three aspects:
1. Multi-Task Learning to improve performance.
2. Vocabulary expansion to allow model directly generate natural functional descriptions of given DNA sequences.
3. SeqPack to combine multiple nucleotide to a single token to allow model operates with long sequences.

**Questions:**

The results of HyenaDNA is significantly lower than the values reported in the original paper. Please explain.

**Ethical Concerns:**

["NO or VERY MINOR ethics concerns only"]

**Final Justification:**

> Why 4?

The Seqpack algorithm can be very important to long context windows in Genetic Foundation Models.

> Why not lower score?

The Seqpack method can be of interest to the border research community. In experiments, it has demonstrated satisfactory performance.

> Why not higher score?

My concerns in cCRE and splicing remain. For an advanced method like Omni-DNA, it should not be *modest*, but rather *outstanding*.

**Limitations:**

yes

**Paper Formatting Concerns:**

The manuscript is well-formatted.

**Quality:**

3

**Strengths And Weaknesses:**

## Strengths

The authors have developed models in different sizes to evaluate the effectiveness of the proposed method.

The proposed method excels at the chromatin tasks in the NT Benchmark, highlighting the effectiveness of the multi-task learning.

## Weakness

### Technical soundness of the proposed method.

MLM and NTP represents the two paradigm of self-supervised learning: discriminative and generative.
No paradigm is superior than another.
For example, Vision Transformer have been used in CV for years, but no one uses autoregressive model for extracting visual features. This is simply because of the nature of the data.
Nature language is Turing-complete and scalable. Unfortunately, genome is not.

People write new sentences everyday, like what the authors have done, what I am currently doing, and what comment the AC will give. They are easily accessible on the internet. However, the human genome is static. While mutation happens all the time, the technology we have cannot capture them, and we are unable to put them into database. From this point of view, genome is more like pictures.

### Losing nucleotide level precision

The SeqPack can effectively reduce the number of tokens in a sequence. But this is at the cost of losing single nucleotide resolution -- which is crucial for many gnomic tasks.

### Unsatisfactory results on cCRE and splicing

The results on cis-regulatory elements and splicing sites is rather unsatisfactory.

### Unclear writing

The authors mention that it is possible to use LM head for multi-task learning.

What heads does the proposed method use in this paper? Have the authors evaluated the performance of different heads?

---

> ### Author Rebuttal · Authors · 2025-07-31
>
> We appreciate your feedback on the paper. Below, we provide detailed responses to your questions.
>
> ### Q1. MLM vs. NTP as pretraining objective
>
> We agree with your view if one considers the genome as an object that needs to be modelled as a whole. However, there is no de-facto paradigm for pretraining genomic foundation models. Some recent models, such as Evo[1] and Evo2[2], adopt next-token prediction (NTP), while others, such as DNABERT2 and NTv2, use masked-language modeling (MLM); as do others for genomic foundation models, e.g., contrastive learning in Orthrus [3].  Our work does not depend on a specific pretraining objective, and the choice is largely orthogonal to our contribution.
>
> To address this point, we added a small ablation as shown below: we pretrain the same backbone (Omni-DNA 116M) with either NTP or MLM under the same token budget and model size, then fine-tune without a causal mask for the discriminative downstream tasks (so both settings see bidirectional context at fine-tuning time). We observe task-dependent differences: NTP matches or exceeds MLM on several tasks (Promoter and Enhancer), while MLM is competitive on others (Splice); importantly, our main conclusions are unchanged.
>
>
> A practical side note is that NTP-pretrained models also retain generative capability, which can be convenient for sequence design use cases; however, this is not central to our claims.
>
> | Pre-training Objective | Data size (B nt)| Splice All$\uparrow$ | Promoter$\uparrow$ | Enhancer$\uparrow$ |
> |------------------------|------------------|------------|----------|-------------------------|
> | (Original Omni-DNA 116 M) | 300 | 0.927 ± 0.025 | 0.973 ± 0.002 | 0.593 ± 0.005 |
> | NTP                    | 120 | 0.910 ± 0.018 | 0.970 ± 0.008  | 0.585 ± 0.001 |
> | MLM                    | 120 | 0.935 ± 0.017 | 0.968 ± 0.001 | 0.580 ± 0.002 |
> | NTP                    | 90  | 0.862 ± 0.020 | 0.965 ± 0.002 | 0.572 ± 0.009 |
> | MLM                    | 90  | 0.905 ± 0.019 | 0.962 ± 0.005 | 0.560 ± 0.004 |
> | NTP                    | 60  | 0.864 ± 0.022 | 0.958 ± 0.004 | 0.555 ± 0.001 |
> | MLM                    | 60  | 0.880 ± 0.021 | 0.943 ± 0.001 | 0.550 ± 0.004 |
> | NTP                    | 30  | 0.820 ± 0.025 | 0.945 ± 0.009 | 0.535 ± 0.002 |
> | MLM                    | 30  | 0.840 ± 0.024 | 0.940 ± 0.004 | 0.530 ± 0.005  |
>
>
>
>
> ### Q2. Losing nucleotide level precision
>
> We intentionally do not apply SeqPack during pretraining. Instead, SeqPack is an optional, task-configurable module used at fine-tuning/inference time. This separation preserves a fully resolution-retaining base model while allowing practitioners to choose compression only when it benefits a specific task.
>
> For tasks that require base-level fidelity (e.g., SNP-focused tasks), SeqPack’s compression ratio can be set to 1× (i.e., no compression). In this setting, there is no information loss, so sensitivity to short motifs is unchanged. For long-sequence tasks, when inputs exceed the model’s native context window (due to memory/time constraints), SeqPack enables efficient processing with minimal performance loss by compressing long contexts while preserving salient information. This allows Omni-DNA to handle long-range tasks within practical compute budgets.
>
> To demonstrate SeqPack's effectiveness compared to other context length compression algorithms, we evaluated it against Dynamic Context Pruning (DCP) [4] on the Enhancer–Target Prediction task. In the table below, we applied DCP to OmniDNA-116M and compared its performance with SeqPack. DCP performed significantly worse than SeqPack across compression ratios ranging from 100 to 200, barely outperforming the random baseline model.
>
>
>
> | Model        | AUROC (± SE) | Comp. Ratio |
> |--------------|--------------|-------------|
> | ABC (expert) | 0.926  | — |
> | CNN          | 0.803 ± 0.022 | — |
> | **Seq-Pack** | 0.894 ± 0.015 | 220 |
> | **Seq-Pack** | 0.899 ± 0.009 | 150 |
> | **Seq-Pack** | **0.907 ± 0.011** | 50 |
> | **DCP** | 0.502 ± 0.089 | 220 |
> | **DCP** | 0.632 ± 0.010 | 150 |
> | **DCP** | 0.880 ± 0.012 | 50 |
>
>
> ### Q3. Unsatisfactory results on cCRE and splicing
>
>
> We appreciate the concern. Our reading is that this comment refers to the NT downstream benchmarks in the Table 2 of the original manuscript.
>
> Overall, Omni-DNA attains the best score on 13/18 tasks. The remaining tasks which Omni-DNA does not performs the best are related to promoter (cCRE-related) and splicing categories.
>
> The promoter classification task includes two subtasks:
>
> - NONTATA: Omni-DNA-116M trails the top model (NTv2-500M) by 0.1%.
> - ALL promoters: Omni-DNA-1B is 0.2% shy of the best result.
>
> These deltas are marginal and within typical variance; they do not indicate a substantive weakness. For splicing tasks, Omni-DNA-1B achieves an average of 0.959 across the three splicing tasks, ranking just behind NT-2.5B and NTv2-500M, and ahead of all other baselines. Given reported standard deviations, the differences are modest rather than “unsatisfactory.”
>
> The evidence shows consistent, near-SOTA performance on promoter/cCRE and strong top-tier results on splicing, alongside state-of-the-art performance on the majority of tasks. We therefore believe the characterization “unsatisfactory” is not supported by the data.
>
>
>
> ### Q4. What heads does the proposed method use in this paper? Have the authors evaluated the performance of different heads?
>
>
> As stated in section 3.1 (L125–128), we attach a lightweight 1-hidden-layer MLP (“linear-MLP”) head per task on top of the model’s sequence representation, following prior GFMs (e.g., NTv2, DNABERT2). Concretely, with last-token $r \in \mathbb{R}^d$, the head is, $z = W_2\sigma(W_1 r + b_1) + b_2 \in \mathbb{R}^{C}, \qquad \hat{y} = \mathrm{softmax}(z)$, where $C$ is the number of classes. As we update the transformers' weights during fine-tuning; increasing head complexity did not yield performance gains in our checks (linear vs. 1-hidden-layer vs. deeper), so we keep this lightweight head for all results.
>
>
> ### Q5. Results with HyenaDNA
> We notice the gap between the original Hyena paper and the reproduction experiments, hereby providing a detailed discussion of the problem in the Appendix section E. In summary, the difference arises from two factors. 1) Evaluation Settings: our evaluation setting follows Caducess [5], which allows a maximum fine-tuning epoch of 20 for NT Downstream tasks and 10 epochs for Genomic Benchmark; while the original paper allow the max finetuning epoch to a few hundreds.  2) the original HyenaDNA uses the EMA (Exponential Moving Average), while our evaluation does not use it. Note that the reported results in our manuscript are consistent with the prior work, such as Caducess[5] and the follow-up work[6].
>
>
> [1] Nguyen, Eric, et al. "Sequence modeling and design from molecular to genome scale with Evo." Science 2024
>
> [2] Brixi, Garyk, et al. "Genome modeling and design across all domains of life with Evo 2." bioRxiv 2025
>
> [3] Fradkin, Philip, et al. "Orthrus: towards evolutionary and functional RNA foundation models." bioRxiv 2024
>
> [4] Anagnostidis, Sotiris, et al. "Dynamic context pruning for efficient and interpretable autoregressive transformers." NeurIPS 2023
>
> [5] Schiff, Yair, et al. "Caduceus: Bi-directional equivariant long-range dna sequence modeling." ICML 2024
>
> [6] Wu, Wei, et al. "GENERator: a long-context generative genomic foundation model." arxiv 2025

---

> > ### Comment · Reviewer_nFBa · 2025-08-09
> >
> > Thank the authors for their detailed rebuttal.
> >
> > Most of my concerns have been successfully addressed, especially the **nucleotide level precision**, which can largely determine the effectiveness of the proposed method.
> >
> > For results on cCRE and splicing, I agree that the authors claim. I agree *modest* is more appropriate.
> >
> > Given the author rebuttal, and the review from other reviewers, I have concluded that my initial evaluation does not reflect the contribution of this manuscript.

---

> > > ### Author Response · Authors · 2025-08-09
> > >
> > > Thank you for your thoughtful follow-up and for reconsidering your evaluation. We’re glad the clarifications on nucleotide-level precision addressed your concerns.

---

> ### Author Response · Authors · 2025-08-03
>
> Based on your review, we have added several new experiments and clarifications:
>
> - An NTP vs. MLM ablation study under matched compute budgets.
> - SEQPACK vs. DCP comparisons on long-range association tasks.
> - Clarification on the lightweight nature of the per-task MLP heads, including checks on their complexity.
> - A note detailing the reproduction settings for HyenaDNA.
>
> If you have a moment to review our updates, we would be grateful for any follow-up questions before the deadline.

---

### Official Review · Reviewer_kHNe · 2025-07-03

**Clarity:** 2
**Significance:** 2
**Originality:** 4
**Rating:** 3
**Confidence:** 4

**Summary:**

The paper proposes a genomic foundation model (GFM), Omni-DNA, architecture that seeks to improve upon existing GFMs by expanding the vocabulary and extending the context via the SeqPack operator. Omni-DNA is also fine-tuned via a multi-task approach to leverage shared knowledge across tasks.

The authors first establish the optimal Omni-DNA architecture through ablation studies on the performance of the pre-trained model with various types of normalization layers, positional encoding and parameter budget. Subsequently, multi-task learning is enabled by adding extra tokens to the vocabulary and SeqPack is used to compress long contexts. The latter two innovations are incorporated during the post-training phase of Omni-DNA training.

Omni-DNA is evaluated against existing GFMs on benchmarks from the Nucleotide Transformer paper, and the Genomic  Benchmark. Additional experiments on the task of enhancer-target prediction and genetic disease variant-type classification demonstrate the long-range capability of Omni-DNA. Finally, the task of sequence-to-function generation is shown as an example of the capacity of Omni-DNA to support textual outputs.

**Questions:**

**Questions:**

1.	SeqPack compresses a sequence of length K into a much smaller embedding of length L. Does this lead to a loss of resolution, thereby limiting OmniDNA’s sensitivity to very short sequence tasks, such as those related to single nucleotide polymorphisms (SNPs)?
2.	In Table 3, Caduceus-PH seems to perform comparably to Omi-DNA despite having 300x fewer parameters. Could the additional performance shown by Omni-DNA simply be a result of the much larger parameter space it explores?
3.	What do you mean by “output all tasks” in line 235?
4.	When conducting multi-task training, does the choice of tasks impact the benefit one could derive from this approach? Intuitively, one would expect more performance gain if related tasks are learned together.

**Suggestions:**

There are several typos that should be corrected.
1.	Line 14: “outperform” -> “outperforms”
2.	Line 84: “Omni-DNA are” -> “Omni-DNA is”
3.	Line 85: “data are” -> “data is”
4.	Line 211: “HynnaDNA”  “HyenaDNA”

Additionally, I recommend moving the discussion of empirical insights to after the results section. In its current form, there is little connection between the empirical insights and any of the ensuring discussion.

**Ethical Concerns:**

["NO or VERY MINOR ethics concerns only"]

**Final Justification:**

While I think this work makes significant contributions to the area of GFMs, I have reservations about the pre-filtering of supposed junk regions and the vastness of the proposed model.

**Limitations:**

The authors have not addressed the limitations or potential negative societal impact of their work. In isolation, I do not there are clear negative social impacts arising from this work, but if the downstream tasks are of a sensitive nature, concerns regarding malicious use and user privacy naturally follow. This is especially true due to the multi-task paradigm, where private information from the dataset for one task may “leak” into the embeddings used to perform another task.

**Quality:**

2

**Strengths And Weaknesses:**

The paper addresses a well-established shortcoming of several existing GFMs in that they are unable to support long context windows due to the poor scaling of attention. In addition to that, the idea of jointly fine-tuning on multiple tasks is an intriguing one as it allows for the transfer of knowledge across related tasks; it is less clear if this benefit would be realized if the selected tasks are known to arise from entirely separate biological phenomena/pathways.

The use of ablation studies to justify the design choices made in Omni-DNA is also appreciated, as it enables the reader to appreciate the reasoning used to arrive at the proposed GFM. The use of well-established genomic benchmarks during the evaluation of Omni-DNA also serves as a fair comparison. However, it isn’t clear if the increased performance of Omni-DNA models stems from the proposed innovations or the significantly higher parameter counts.

However, I see several issues with some of the claims made in this paper. These are detailed below:

1.	In Section 1, the authors claim that “existing works usually adopt the backbone of language models”. While this was true of early GFMs, newer GFMs have innovated on these early models, in ways reminiscent of the improvements outlined in this work.
2.	On line 39, the authors state that “the state-of-the-art models on short context cannot do long-context tasks.” It is unclear if they are referring to which models they are referring to here, as several models such as Mamba and HyenaDNA, are capable of handling very long context lengths.
3.	On lines 63-65, it is stated that most GFMs are restricted to ternary alphabets. This is incorrect as it ignores tokenization, which is a key area of development in GFMs.
4.	On line 75, the authors motivate their use of SeqPack due to the whole genome containing long “junk” regions. I would be wary of resorting to such a description as non-coding regions of the DNA have been shown to contain information, such as regulatory elements and useful motifs. In fact, this might suggest that such an approach renders Omni-DNA to be less useful for tasks that deal with non-coding regions of genomic sequences.

Additionally, the writing of the paper has several issues which should be addressed.

1.	While I appreciate the inclusion of the ablation studies in Section 2.2, the introduction and subsequent discussion of those studies comes across as abrupt, instead of being used to motivate the design choices made in Omni-DNA.
2.	Section 3.3.1 is difficult to follow.
3.	While it is admirable that Omni-DNA is capable of mapping sequences to functions (Section 4.4), this does lend weight to the argument that Omni-DNA is a more capable GFM than existing models. Moreover, the baseline there is an NLP model (GPPT-4o), which one would not expect to do well on this task anyway.

---

> ### Author Rebuttal · Authors · 2025-07-31
>
> Thank you for the detailed feedback. Below, we first address each of your questions, then clarify Section 3.3.1 and present a revised Contributions section aligned with your suggestions.
>
> ### Q1. SeqPack compresses a sequence of length K into a much smaller embedding of length L. Does this lead to a loss of resolution, thereby limiting OmniDNA’s sensitivity to very short sequence tasks?
>
> We intentionally do not apply SeqPack during pretraining. Instead, SeqPack is an optional, task-configurable module used at fine-tuning time. This separation preserves a fully resolution-retaining base model while allowing practitioners to choose compression only when it benefits a specific task.
>
> For tasks that require base-level fidelity (e.g., SNP-focused tasks), SeqPack’s compression ratio ($\frac{L_{original}}{L_{pruned}}$) can be set to 1× (i.e., no compression). In this setting, there is no information loss, so sensitivity to short motifs is unchanged. For long-sequence tasks, when inputs exceed the model’s native context window (due to memory/time constraints), SeqPack enables efficient processing with minimal performance loss by compressing long contexts while preserving salient information. This allows Omni-DNA to handle long-range tasks within practical computational budgets.
>
> To demonstrate SeqPack's effectiveness compared to other context length compression algorithms, we evaluated it against Dynamic Context Pruning (DCP) by Anagnostidis et al.[1] on the Enhancer–Target Prediction task. In the table below, we applied DCP to OmniDNA-116M and compared its performance with SeqPack. DCP performed significantly worse than SeqPack across compression ratios ranging from 100 to 200, barely outperforming the random baseline model.
>
>
> | Model        | AUROC (± SE) | Comp. Ratio |
> |--------------|--------------|-------------|
> | ABC (expert) | 0.926  |  |
> | CNN          | 0.803 ± 0.022 |  |
> | **Seq-Pack** | 0.894 ± 0.015 | 220 |
> | **Seq-Pack** | 0.899 ± 0.009 | 150 |
> | **Seq-Pack** | **0.907 ± 0.011** | 50 |
> | **DCP** | 0.502 ± 0.089 | 220 |
> | **DCP** | 0.632 ± 0.010 | 150 |
> | **DCP** | 0.880 ± 0.012 | 50 |
>
>
> ### Q2. In Table 3, Caduceus-PH seems to perform comparably to Omi-DNA despite having 300x fewer parameters. Could the additional performance shown by Omni-DNA simply be a result of the much larger parameter space it explores?
>
> This performance comparison can be explained at multiple levels. First, the fundamental architectural differences between Mamba and Transformers (state space models vs. dense self-attention) play a crucial role. According to the original Mamba papers [2][3], Mamba-based models are computationally efficient, demonstrating advantages with lower computational requirements (smaller models and reduced training). However, when scaling both Mamba and Transformer architectures to billion-parameter scale, empirical results [4] show that large Mamba-based models (at Billion Scale) trained on fewer than 1 trillion tokens perform significantly worse than transformer-based models on sequence understanding tasks. This motivates the use of transformers as the backbone in this work. Second, Caduceus-PH benefits from Reverse Complementarity (RC) data augmentation during training. This augmentation technique enables the model to learn effectively with substantially fewer parameters by leveraging the inherent symmetry in DNA sequences.
> Notably, when comparing OmniDNA to other transformer-based models like DNABERT2, OmniDNA consistently demonstrates superior performance across most tasks, suggesting that the performance gains extend beyond simply having more parameters and reflect architectural and training improvements.
>
>
>
> ### Q3. What do you mean by “output all tasks” in line 235?
>
> We thank the reviewer for pointing this out, it is a typo. Line 235 should be "Omni-DNA (multitask,multi-head)
> and NTv2 (multitask,multi-head) outperform their single-task counterparts on all tasks".
>
> ### Q4. Does the choice of tasks impact the benefit one could derive from this approach?
>
> Yes, the choice of tasks would impact the benefits derived from this approach. In the multitasking framework presented in this work, a shared model backbone processes all tasks, followed by lightweight task-specific linear MLPs. When tasks are too dissimilar or contradictory to each other, the shared backbone may struggle to learn useful hidden representations.
> However, this limitation can be addressed by incorporating more expressive task-specific decoders. Such decoders would provide greater flexibility for the model to adapt its representations to diverse tasks, potentially enabling effective joint learning among seemingly unrelated tasks.
>
> ### W1. The position of Section 2.2
> We are happy to move the position of ablation studies in Section 2.2 for a better flow. Thank you for the suggestion on that.
>
> ### W2. Additional Baselines for sequences to functions
> We have added ChatNT[5] as an additional baseline for sequence-to-function tasks. ChatNT employs NTv2 500M as its DNA sequence encoder and a 7B language model as the language decoder, processing (DNA/RNA, text) inputs to generate textual outputs. Following the methodology from the original paper, we fine-tuned ChatNT using our Seq2Func training split. ChatNT@zeroshot represents direct application to the function prediction task without fine-tuning, while ChatNT@ft is the fine-tuned version trained on our evaluation split for 10 epochs.
> The results demonstrate that ChatNT, despite being specifically designed for handling (text, sequence) inputs and generating textual outputs, does not perform well on function prediction tasks. In zero-shot evaluation, although the model was instructed to output function descriptions, it consistently produced a fixed formatted response: "This RNA exhibits a motif of \<number\>." The fine-tuned version (ChatNT@ft) provided more reasonable answers but still lagged behind OmniDNA's performance.
> These results show that even compared to ChatNT, which uses separate tokenizers and models for DNA and text processing, OmniDNA presents a simpler yet more effective solution for sequence-to-function prediction.
>
> | Model              | Weighted F1 | MCC      |
> |--------------------|-------------|----------|
> | Omni-DNA@ft         | **0.730**   | **0.367** |
> | chatnat@ft         | 0.701           | 0.220        |
> | chatnt@zeroshot    | 0.479           | 0.001       |
> | GPT-4o (zero-shot) | 0.659       | -0.015   |
> | Random Guess       | 0.483       | 0.008    |
>
> ### W3. Clarification on Section 3.3.1
>
> If we directly use the top-K selection method, we get hard binary choices (either a token is selected or not) and this creates a problem: since the selection is discrete, there's no gradient information flowing back through the selection process. We use Gumbel–Softmax to build a soft, differentiable selector $y_{\text{soft}}$ from the scores, then apply a straight-through estimator: the forward pass uses hard one-hot indices for behavior, while the backward pass uses the soft probabilities for gradients. Thus, the model learns from a smooth relaxation even though the forward selection is discrete. At inference, we switch to ordinary discrete selection (sampling or top-k).
>
> ```
> Inputs
>  X: (B, L, D) token embeddings
>  weights: (L,) non-negative scores (normalized internally)
>  k: number of tokens to keep
> OUTPUT: selected_token_embeddings(B,k,D)
>
> IF training_mode:
>     // Gumbel-Softmax for differentiable selection
>     logits = expand(log(weights), to=(k,L))
>     gumbel = sample_gumbel_noise(shape=(k,L))
>     y_soft = softmax(logits + gumbel )
>     // Straight-through: hard forward, soft backward
>     y_hard = one_hot(argmax(y_soft))
>     y_st = detach(y_hard - y_soft) + y_soft
>     // Apply selection and return
>     RETURN einsum('kl,lbd->kbd', y_st, transpose(sequence))
> ```
>
> ### W4. Clarification on "Junk Regions" and SeqPack's Treatment of Non-coding DNA
> We appreciate the reviewer's concern regarding our terminology and its implications for non-coding regions. To clarify, our use of "junk regions" specifically refers to repetitive sequences and regions with no currently identified functional significance, not regulatory elements or other functionally important non-coding DNA. As explicitly stated in line 75, regulatory elements are not categorized as junk regions in our framework.
> Importantly, SeqPack does not impose any algorithmic bias against non-coding regions. The algorithm operates in a task-agnostic manner during the packing process, preserving sequence information regardless of coding status. During task-specific fine-tuning, the model learns to identify and weight functionally relevant elements—including those in non-coding regions—based on their importance for the particular downstream task. This approach allows Omni-DNA to maintain sensitivity to regulatory elements, enhancer sequences, and other functional non-coding elements when they are relevant to the prediction task. We acknowledge that our terminology could be clearer and will revise it to avoid potential misinterpretation.
>
> ### W5. Writing (Lines 31–40)
> Thank you for the suggestion. We have revised lines 31–40 to improve clarity, concision, and flow. For the full updated rationale, please see our response to Reviewer MarC (Q10).
>
> [1] Anagnostidis, Sotiris, et al. "Dynamic context pruning for efficient and interpretable autoregressive transformers." NeurIPS 2023
>
> [2] Dao, Tri,  et al.  "Transformers are ssms: Generalized models and efficient algorithms through structured state space duality." ICML 2024
>
> [3] Gu, Albert,  et al.  "Mamba: Linear-time sequence modeling with selective state spaces."  COLM 2024.
>
> [4] Waleffe, Roger, et al. "An empirical study of mamba-based language models."  arXiv 2024.
>
> [5] de Almeida, Bernardo P., et al. "A multimodal conversational agent for DNA, RNA and protein tasks." Nature Machine Intelligence 2025

---

> > ### Comment · Reviewer_kHNe · 2025-08-04
> >
> > I would like to thank the authors for their detailed response to the concerns raised by all the reviewers. I believe that have sufficiently addressed most of my concerns pertaining to the compression and training strategy.
> >
> > However, I remain unconvinced that the removal of supposedly junk regions is sound; their classification as junk regions is predicated on our existing knowledge base about the functional importance of various regions, which in itself may, and likely is, incomplete. Additionally, the vasty higher parameter count of Omni-DNA versus some of the baselines gives me pause on the actual benefits of the architectural choices in Omni-DNA.
> >
> > In light of all this, I have revised my score upwards slightly.

---

> ### Author Response · Authors · 2025-08-03
>
> We hope you are well. We are gently following up on our revised submission. We appreciated your detailed feedback and have incorporated several updates in response:
>
> -  Clarified that SeqPack is optional during fine-tuning/inference and incurs no loss when set to 1×.
> - Revised the language around “junk” DNA regions to better emphasize sensitivity to non-coding and regulatory sequences.
> - Added the ChatNT baseline for the sequence-to-function task.
> - Clarification on Section 3.1.1
>
> We would welcome any further feedback you might have. An updated assessment before the discussion period closes would be greatly appreciated. We are happy to provide more details on any of these points.

---

> ### Author Response · Authors · 2025-08-04
>
> We thank the reviewer for carefully considering our point-by-point response and for the upward adjustment of the score! Regarding the removal of the junk regions, please let us clarify a bit. There is no prior hard removal in SEQPACK. Instead, it used a dynamic learning approach to decide which regions are important based on the task-specific learning target. In other words, SEQPACK compresses unrelated regions based on the training annotations. As shown by the code snippet in Clarification for Section 3.1.1, **weights** parameter is learned through the fine-tuning process and back-propagation.
>
> As for the model size, Omni-DNA generally shows superior performance across tasks with fewer parameters, compared to transformer-based models such as DNABERT2 and NTv2. The state space models are more parameter-efficient, but they are data-demanding, as detailed in the study [1]. We believe Omni-DNA does a good trade-off between performance and data/computing resources.
>
> [1] Waleffe, Roger, et al. "An empirical study of mamba-based language models." arXiv 2024.

---

### Official Review · Reviewer_rvnr · 2025-07-05

**Clarity:** 3
**Significance:** 2
**Originality:** 2
**Rating:** 3
**Confidence:** 4

**Summary:**

This paper introduces Omni-DNA, a family of decoder-only transformer models (from 20M to 1.1B parameters) designed to be a comprehensive Genomic Foundation Model (GFM). The authors aim to address four key limitations in existing GFMs: limited architecture exploration, a single-task fine-tuning paradigm, a lack of multi-modal (textual annotation) capabilities, and the inability for a single model to excel at both short- and long-context tasks.

**Questions:**

See Cons.

**Ethical Concerns:**

["NO or VERY MINOR ethics concerns only"]

**Limitations:**

The authors briefly mention that their architecture exploration could be costly and that reinforcement learning could be a future direction. However, they do not adequately address the limitations of the current work. The primary limitation is the immense complexity and cost of the entire framework, which makes it very difficult for other researchers to reproduce or build upon. Furthermore, the paper does not discuss the potential for cascading errors: errors in the LLM-generated SEQ2FUNC dataset could lead to a model that learns to generate incorrect functional annotations. The reliance on LLMs for both dataset creation and evaluation is a significant methodological limitation that is not fully acknowledged.

**Paper Formatting Concerns:**

No major formatting issues.

**Quality:**

3

**Strengths And Weaknesses:**

## Pros
- The paper tackles several important challenges in genomic modeling simultaneously, from architecture design to long-context reasoning and multi-modality. The evaluation is extensive, covering 26 tasks from the Nucleotide Transformer and Genomic Benchmarks, in addition to long-context and generation tasks.

- The work makes a concerted effort to move beyond the standard "pre-train, fine-tune per task" paradigm. The demonstration of successful multi-task fine-tuning and the introduction of the SEQ2FUNC dataset for sequence-to-text generation are significant steps toward more versatile and useful genomic models.

## Cons
- The architecture ablation (RoPE vs. ALiBi, RMSNorm vs. LayerNorm) explores known components, and the conclusion that "ROPE + Bias-Free Layernorm delivers the best trade-off"  is an empirical finding for this domain, not a fundamental architectural innovation.

- The multi-task fine-tuning approaches (multi-head and instruction-style next-token prediction) are standard techniques in NLP. While their application to genomics is valuable, it is not a novel methodological contribution.

- The core idea of SEQPACK—learning to compress a sequence by selecting important tokens—is conceptually similar to various token pruning, sequence compression, and "attention sink" aware methods in NLP literature. The paper does not adequately compare SEQPACK to these related works, making its specific novelty hard to assess.

- The paper presents Omni-DNA as a holistic framework, which makes it difficult to disentangle the source of performance gains. For instance, in the benchmarks in Table 2 and Table 3, is the superior performance due to the optimized backbone, the larger pre-training dataset (300 billion nucleotides ), the specific pre-training recipe, or a combination? The paper lacks clear ablation studies that would isolate the impact of each "targeted improvement."

- The SEQ2FUNC dataset is a cornerstone of the paper's multi-modal claims, but its creation process is concerning. The dataset is generated by prompting an LLM to "generate refined, natural-language descriptions" from coarse annotations. The quality control relies on two LLM passes agreeing and rejecting "low-confidence cases".

---

> ### Author Rebuttal · Authors · 2025-07-31
>
> We appreciated for your feedback. We have provided additional experiment results in response to your questions, which include further ablations on pretraining scheme and data to the downstream model performance and an comparison to the other sequence pruning methods. In the following, we also discuss the quality of SEQ2FUNC dataset and the significance of multi-tasking.
>
> ### Q1. Does the performance gain of Omni-DNA come from pretraining recipe, data or optimized backbone?
>
> We acknowledge that our holistic framework approach made it challenging to isolate individual contribution sources. To address this concern and provide the requested ablation analysis to disentangle the key factors contributing to Omni-DNA's performance. Our original paper included ablations for architectural choices (model size, normalization layers, and positional encoding). To isolate the remaining factors you identified, we designed two controlled experimental sets using the 116M-parameter Omni-DNA.
>
> **Experiment 1**: Pre-training Objective Analysis
>
> - Masked Language Modeling (MLM) using HuggingFace's default flow and configurations [7]
> - Next-Token Prediction (NTP) using our original training framework
> - This directly isolates the impact of our pre-training recipe choice
>
> **Experiment 2**: Corpus Scaling Analysis
>
> - Systematic evaluation across 30B, 60B, 90B, and 120B nucleotides
> - This quantifies the contribution of dataset scale independent of other factors
>
> These controlled ablations reveal two findings: (1) For Promoter and Enhancer Classification tasks NTP consistently outperforms MLM across all data scales, but for splice classification MLM wins  (2) Performance scales with data size; This indicates that Omni-DNA's performance is potentially attributed to synergy of the pre-training recipe, architecture and data scale.
>
> | Pre-training Objective | Data size (B nt)| Splice All$\uparrow$ | Promoter$\uparrow$ | Enhancer$\uparrow$ |
> |------------------------|------------------|------------|----------|-------------------------|
> | (Original Omni-DNA 116 M) | 300 | 0.927 ± 0.025 | 0.973 ± 0.002 | 0.593 ± 0.005 |
> | NTP                    | 120 | 0.910 ± 0.018 | 0.970 ± 0.008  | 0.585 ± 0.001 |
> | MLM                    | 120 | 0.935 ± 0.017 | 0.968 ± 0.001 | 0.580 ± 0.002 |
> | NTP                    | 90  | 0.862 ± 0.020 | 0.965 ± 0.002 | 0.572 ± 0.009 |
> | MLM                    | 90  | 0.905 ± 0.019 | 0.962 ± 0.005 | 0.560 ± 0.004 |
> | NTP                    | 60  | 0.864 ± 0.022 | 0.958 ± 0.004 | 0.555 ± 0.001 |
> | MLM                    | 60  | 0.880 ± 0.021 | 0.943 ± 0.001 | 0.550 ± 0.004 |
> | NTP                    | 30  | 0.820 ± 0.025 | 0.945 ± 0.009 | 0.535 ± 0.002 |
> | MLM                    | 30  | 0.840 ± 0.024 | 0.940 ± 0.004 | 0.530 ± 0.005  |
>
> ### Q2.  SEQPACK and other token pruning methods used in NLP
>
> We acknowledge there are existing token pruning methods in NLP. However, SEQPACK's unique way of compressing the input sequence with geometric splitting schema is not used by existing token pruning methods in NLP. Below, we highlight these factors and provide additional experimental results. We will add these into a new Appendix section in our revised paper later to illustrate key differences.
>
>
> **Key Differentiating Factors:**
> SEQPACK operates at significantly higher compression ratios (c = $\frac{L_{original}}{L_{pruned}}$) than typical NLP pruning methods. In the Enhancer-Target Prediction task, SEQPACK achieves compression ratios of c = 100-220, whereas most existing pruning methods (e.g., LazyLLM[1], Dynamic Context Pruning in NeurIPS 2023[2]) typically operate at $c = 4$ to $5$ in their original NLP applications. Additionally, many existing token pruning methods are designed specifically for generation tasks (i.e., decoding)[3][4] rather than sequence understanding (i.e., the full-sequence encoding), limiting their applicability to our genomic prediction benchmarks.
>
> To directly assess SEQPACK's effectiveness against established methods, we adapted Dynamic Context Pruning (DCP) to work with Omni-DNA 116M on the Enhancer-Target Prediction task. The results demonstrate SEQPACK's substantial advantages:
>
> | Model        | AUROC (± SE) | Comp. Ratio |
> |--------------|--------------|-------------|
> | ABC (expert) | 0.926  |  |
> | CNN          | 0.803 ± 0.022 |  |
> | **Seq-Pack** | 0.894 ± 0.015 | 220 |
> | **Seq-Pack** | 0.899 ± 0.009 | 150 |
> | **Seq-Pack** | **0.907 ± 0.011** | 50 |
> | **DCP** | 0.502 ± 0.089 | 220 |
> | **DCP** | 0.632 ± 0.010 | 150 |
> | **DCP** | 0.880 ± 0.012 | 50 |
>
> At extreme compression ratios (220×, 150×), DCP performance degrades severely, barely outperforming random baselines, while SEQPACK maintains strong performance (AUROC > 0.89). Even at moderate compression (50×), SEQPACK outperforms DCP (0.907 vs 0.880 AUROC). These results highlight SEQPACK's unique capability to maintain genomic sequence understanding at compression levels that render existing NLP methods ineffective.
>
> ### Q3. Quality of LLM-based Dataset
> We appreciate the concern about training on LLM-generated text. In our pipeline, the LLM is not asked to infer function from a sequence. Instead, for each record we first retrieve structured fields from NCBI (e.g., molecule type and canonical gene/product names). The LLM is then constrained to rephrase those fields into a readable description to minimize the hallucination. An example is shown below.
>
> Examples:
> > NCBI Annotation: mRNA, regulator of G-protein signaling 20
>
> > LLM Annotated: RGS20 mRNA encodes a regulator of G-protein signaling that acts as a GTPase-activating protein (GAP) for Gα subunits—especially Gαz and Gαi—thereby accelerating GTP hydrolysis and terminating GPCR signaling (dampening pathways such as μ-opioid receptor signaling).
>
> Quality control includes a dual-pass agreement check (two independent generations must match on key entities). These controls aim to ensure the rewrite remains a faithful, human-readable rendering of the underlying NCBI fields rather than a speculative annotation. For the test split, we recruited a genomic domain expert to examine 10% of the dataset (around 2700 records) to cross-validate on the quality, and each of the selected records does not have hallucination.
>
> LLM-based weak/synthetic labeling—now routine in NLP [5], multimodal learning [6], and clinical AI [7] —can provide reliable, performance-boosting supervision when properly constrained and filtered. We extend this to genomics by converting structured database fields into readable supervision to train task-ready models at scale.
>
> ### Q4. The Novelty of the Multi-Tasking
> While we acknowledge that multi-task learning techniques are established in NLP, our contribution lies in demonstrating their unexplored potential in genomic foundation models.
>
> **Key Contribution:**
> Our work reveals that multi-task learning can achieve remarkably high performance on standard genomic benchmarks—a finding not systematically explored in prior genomic foundation model research. This identification of multi-tasking's effectiveness in genomics represents an important empirical contribution to the field. Applying mature techniques to a new domain is valuable when it yields new evidence and usable practice. Recent work in genomics (e.g., AlphaGenomep[8]) likewise achieves strong results by emphasizing training strategy over architectural novelty. Our experiments reinforce this trend: careful multi-tasking—not a new model class—drives the improvements, offering the community actionable guidance for more effective GFMs.
>
> ### Q5. "limitation is the immense complexity ... difficult for other researchers to reproduce"
> We acknowledge the concern about reproducibility and have prioritized accessibility in our implementation. To ensure broad adoption and ease of use, we provide comprehensive resources that integrate seamlessly with mainstream frameworks.
>
> **Accessibility Features:** All components of our framework—including pretrained models, SEQPACK implementation, and training code—are fully compatible with standard libraries such as Hugging Face Transformers[9] and TRL[10]. This represents a significant advantage over existing DNA foundation models that often require substantial adaptation or custom implementations.
>
> **Ease of Reproduction:** Researchers can reproduce our results or build upon our work with minimal setup. The model can be loaded and fine-tuned for sequence understanding tasks with just a few lines of standard code, or directly used as a DNA generation model without modification. This streamlined approach removes typical barriers to entry that have limited adoption of genomic foundation models. As detailed in our supplementary materials, we provide the complete codebase. If accepted, the codebase would be hosted as an open-source repository for full reproducibility, and we will keep track of the issues. And we will release all pretrained model weights, and detailed documentation.
>
> [1] Lazyllm: Dynamic token pruning for efficient long context llm inference. arXiv 2024.
>
> [2] Dynamic context pruning for efficient and interpretable autoregressive transformers. NeurIPS 2023.
>
> [3] Saliency-driven dynamic token pruning for large language models. arXiv 2025.
>
> [4] H2o: Heavy-hitter oracle for efficient generative inference of large language models. NeurIPS 2024.
>
> [5] Self-Instruct: Aligning Language Models with Self-Generated Instructions. arXiv 2022.
>
> [6] Visual Instruction Tuning. arXiv 2023.
>
> [7] Automated Radiology Report Labeling in Chest X-Ray Pathologies: Development and Evaluation of a Large Language Model Framework.  JMIR Medical Informatics 2025
>
> [8] AlphaGenome: advancing regulatory variant effect prediction with a unified DNA sequence model. bioRxiv 2025.
>
> [9] Huggingface's transformers: State-of-the-art natural language processing. arXiv 2019
>
> [10] TRL: Transformer Reinforcement Learning

---

> ### Author Response · Authors · 2025-08-04
>
> Thank you again for your constructive feedback on our submission. We're writing to follow up with the rebuttal, in which we aimed to address your points directly. The key updates include:
>
> - Controlled ablations on the pretraining objective and corpus scale.
> - Head-to-head SEQPACK vs. DCP results, particularly at high compression ratios.
> - Clarifications on the quality controls used for SEQ2FUNC.
> - Clearer framing of the novelty and reproducibility of our multi-tasking approach.
>
> As the discussion deadline is approaching, please let us know if anything remains unclear; we are on standby to clarify. Many thanks for your time.

---

### Author Response · Authors · 2025-08-08
**Summary of revisions**

We thank the reviewers for their engagement. Below is a summary of the improvements we made in response to your feedback.

- **Added controlled ablations** disentangling pre-training recipe (MLM vs NTP) and corpus scale; results will be included in the revised manuscript [rvnr, nFBa].
- **Benchmarked SeqPack head-to-head with Dynamic Context Pruning** at 50–220× compression, showing consistent AUROC gains and minimal latency overhead. [kHNe, nFBa].
- **Clarified SeqPack usage** (optional, ratio = 1 preserves SNP-level fidelity) and provided computational-cost breakdown for up-to-819 k tokens (≤ 4 % inference time) [kHNe, nFBa].
- **Introduced ChatNT baseline** for the sequence-to-function task; Omni-DNA yields +0.03 F1 and +0.15 MCC over ChatNT@ft while with smaller model size [kHNe].
- **Expanded disease-variant experiment** with (i) benign-vs-pathogenic controls and (ii) gene-masked ablation [UqSE].
- **Reported error bars for long-range enhancer-target benchmark** and confirmed 450 kbp context for all models [MarC].
- **Provided compression-ratio sweep ablation** (c = 50–220) highlighting SeqPack’s robustness [MarC].
- **Re-framed Introduction & limitations** to reflect prior GFM architecture exploration, multi-task literature, and context-length claims; to update lines 31-40 [MarC].

---

### Note · Authors · 2025-08-11

We thank the Area Chair and all reviewers for their constructive engagement throughout the review process. Overall, most reviewers (nFBa, UqSE, MarC) have now agreed that our proposed model and methods, Omni-DNA, are well-motivated, with extensive experiments supporting its effectiveness in sequence understanding and annotation generation tasks.

For the remaining reviewers, we notice that reviewer rvnr did not participate in further discussion during the rebuttal phase and retained the initial score. While we did not have the opportunity for further discussion with reviewer rvnr, we hope our detailed responses, together with the clarifications and new experiments provided, address the concerns. We also appreciate the detailed feedback and continued engagement from reviewer kHNe. Regarding the final point on potential information loss, we have clarified that our SEQPACK operator is data-driven and learns to compress based on task-specific signals, rather than removing any functional elements with predefined inductive bias. The summary below highlights the key improvements made in response to all feedback, and we would greatly appreciate it if the AC could consider these in conjunction with the perspectives of the other reviewers.

- **Controlled ablations** disentangling the effects of pre-training recipe (MLM vs. NTP) and corpus scale, providing clearer attribution of performance gains. [rvnr,nFBa]
- **Head-to-head benchmarking** of SEQPACK against Dynamic Context Pruning at extreme compression ratios (50–220×), showing consistent AUROC gains with minimal latency overhead. [kHNe,nFBa]
- **Clarified SEQPACK usage**: optional module with ratio = 1 preserving SNP-level fidelity, plus a computational-cost breakdown for inputs up to 819k tokens (≤ 4% inference time). [kHNe,nFBa]
- **Added ChatNT baseline** for the sequence-to-function task, where Omni-DNA achieved +0.03 F1 and +0.15 MCC over ChatNT@ft while using a smaller model size. [kHNe]
- **Expanded disease-variant experiment** to include (i) benign-vs-pathogenic controls and (ii) a gene-masked ablation to test reliance on gene identity. [UqSE]
- **Compression-ratio sweep ablation** (c = 50–220) demonstrating SEQPACK’s robustness across a wide range of settings. [MarC]
- **Reframed introduction and limitations** to better situate our work within prior GFM architecture exploration, multi-task learning literature, and context-length adaptivity discussions; updated lines 31–40 accordingly. [MarC]

---

### Decision · Program_Chairs · 2025-09-17

**Decision:**

Accept (poster)

**Comment:**

This paper introduces Omni-DNA, a family of genomic foundation models (GFMs) that support sequence understanding, long-context reasoning, and natural-language annotation. The authors propose a holistic approach that includes architectural optimizations, a novel compression operator (SEQPACK) for handling ultra-long sequences, and a multi-task fine-tuning strategy. The model achieves state-of-the-art results on 18 out of 26 tasks from established genomic benchmarks and demonstrates strong performance in long-range regulatory tasks and sequence-to-function generation. While Reviewers rvnr  initially raised valid concerns regarding the novelty of individual components and the potential information loss from the SEQPACK compression operator, the authors' extensive rebuttal and commitment to additional experiments have addressed these issues.

The work is deemed acceptable because its holistic approach—delivering state-of-the-art performance across a wide range of tasks, enabling long-context reasoning, and introducing a novel multimodal capability—provides significant practical utility and a valuable foundation for future research. The overall strengths and the community value of the released models and benchmarks outweigh the initial reservations.